# Spectral signatures of the surface anomalous Hall effect in magnetic axion insulators

Mingqiang Gu [1,6], Jiayu Li [1,6], Hongyi Sun[1], Yufei Zhao [1], Chang Liu[1], Jianpeng Liu [2,3✉], Haizhou Lu[1] & Qihang Liu[1,4,5✉]

The topological surface states of magnetic topological systems, such as Weyl semimetals and axion insulators, are associated with unconventional transport properties such as nonzero or half-quantized surface anomalous Hall effect. Here we study the surface anomalous Hall effect and its spectral signatures in different magnetic topological phases using both model Hamiltonian and first-principles calculations. We demonstrate that by tailoring the magnetization and interlayer electron hopping, a rich three-dimensional topological phase diagram can be established, including three types of topologically distinct insulating phases bridged by Weyl semimetals, and can be directly mapped to realistic materials such as $MnBi_2Te_4$/$(Bi_2Te_3)_n$ systems. Among them, we find that the surface anomalous Hall conductivity in the axion-insulator phase is a well-localized quantity either saturated at or oscillating around $e^2/2h$, depending on the magnetic homogeneity. We also discuss the resultant chiral hinge modes embedded inside the side surface bands as the potential experimental signatures for transport measurements. Our study is a significant step forward towards the direct realization of the long-sought axion insulators in realistic material systems.

[1] Shenzhen Institute for Quantum Science and Engineering and Department of Physics, Southern University of Science and Technology, Shenzhen, China. [2] School of Physical Science and Technology, ShanghaiTech University, Shanghai, China. [3] ShanghaiTech Laboratory for Topological Physics, ShanghaiTech University, Shanghai, China. [4] Guangdong Provincial Key Laboratory for Computational Science and Material Design, Southern University of Science and Technology, Shenzhen, China. [5] Shenzhen Key Laboratory of Advanced Quantum Functional Materials and Devices, Southern University of Science and Technology, Shenzhen, China. [6] These authors contributed equally: Mingqiang Gu, Jiayu Li. ✉email: liujp@shanghaitech.edu.cn; liuqh@sustech.edu.cn

Magnetic topological systems have drawn significant attention recently due to their unconventional bulk transport properties and surface states[1]. The topological properties of these magnetic topological phases are typically described by nontrivial bulk topological indices[2]. On the other hand, the topological nature of the surface states also implies that there would be nontrivial surface transport behavior in these magnetic topological systems, which is closely related to their exotic bulk response properties. Typical examples are three-dimensional (3D) insulators with non-vanishing Chern–Simons orbital magnetoelectric coupling exhibiting effective axion electrodynamics, which are characterized by fractionalized surface anomalous Hall effect[3–6]. If either time-reversal ($T$) or inversion symmetry ($I$) is present, the coupling phase angle $\theta$ must be quantized as 0 or $\pi$ (modulo $2\pi$), the latter of which ($\theta = \pi$) with an energy gap at the surface is also known as "axion insulators" defined in 3D systems[3,4,7]. Thus, the axion insulator exhibits a quantized bulk magnetoelectric coupling coefficient, which is equivalent to a half-quantized surface anomalous Hall conductivity (AHC) $e^2/2h$[3,4].

3D $T$-preserved topological insulators (TIs) possess the $\theta = \pi$ condition. However, the resulting half-quantized surface AHC is totally compensated by the gapless surface Dirac cones of TIs[8]. Therefore, the $T$-preserved axion insulator phase was typically realized by introducing extrinsic magnetic dopants to the top and bottom surfaces of a 3D TI to gap the surface states[9–12], while keeping the bulk still nonmagnetic. On the other hand, bulk magnetic TIs with inversion symmetry are categorized as $I$-preserved axion insulators[13]. Examples include the recently discovered superlattice-like stoichiometric compounds $MnBi_2Te_4/(Bi_2Te_3)_n$ in either ferromagnetic (FM) or antiferromagnetic (AFM) phases[14–21]. Besides, this Van der Waals (VdW) layered material family also provides an ideal platform for realizing fruitful topological phases, such as Chern insulator, quantum spin Hall (QSH) insulator, and high-order TI[22–29].

Compared with the quantum anomalous Hall state where the topological nature is well established by the chiral edge states carrying a nonzero Chern number, the direct evidences of an axion insulator, such as the topological magnetoelectric effect or the resultant surface AHC, are much more challenging to measure[30,31]. Conventional quantum transport measurements inevitably count the top and bottom surface together, giving rise to a (½ + ½) or (½ − ½) quantized AHC, depending on the relative magnetic orientation of the two surfaces. To avoid indistinguishable signature with quantum anomalous Hall effect, a $T$-preserved axion insulator typically requires different magnetic doping to the top and bottom surfaces. In a magnetic hysteresis loop, this slab setup would give rise to two quantum anomalous Hall states with opposite Chern numbers connected by an intermediate insulating phase with zero Hall plateau[32]. In even-layer $MnBi_2Te_4$ slabs, zero Hall plateau is observed as an indirect evidence of the $I$-preserved axion insulator phase[23]. However, such zero Hall plateau can also be presented by a trivial case where both surface gaps are dominated by finite-size effect and thus do not contribute any surface anomalous Hall conductivity at all[33–35]. Therefore, despite being a fascinating theoretical concept, some key issues about the surface AHC of axion insulators, such as the locality and the device design, still remain elusive. More importantly, a clear prediction of the unique signature of axion insulators that can be experimentally detected is still lacking.

In this work, by tuning the interlayer coupling and magnetization of a generic model Hamiltonian, we construct a topological phase diagram with direct mappings to 3D $MnBi_2Te_4/(Bi_2Te_3)_n$ compounds, including axion insulators, Weyl semimetals, 3D Chern insulators, and 3D QSH insulators. Their distinctive surface AHC features are comprehensively studied. In addition to the model study, we then construct atomistic Hamiltonians from density-functional theory (DFT) with close reliance on realistic attributes of materials, yielding a direct comparison with the angle-resolved photoemission spectroscopy (ARPES) measurements for the band dispersions. By projecting the Chern number of a thick slab onto each VdW layer, we find that such real-space, local Chern marker in the axion insulator phase is well localized at the surface and results in a surface AHC either saturated at or oscillating around $e^2/2h$, depending on the magnetic homogeneity. In comparison, the 3D Chern insulator phase, as well as the Weyl semimetal phase, does not manifest a well-defined surface AHC. Remarkably, we propose that the surface anomalous Hall effect in the axion insulator phase of $MnBi_2Te_4/(Bi_2Te_3)_n$ leads to an unusual chiral hinge mode embedded in the side surface Bloch states, which clearly distinguishes from that in a trivial insulator. Such a clear signature is supposed to be detectable by appropriate surface transport measurements.

## Multiple topological phases from model Hamiltonian calculations

To begin with, we tune the hopping parameter between the VdW layers in $MnBi_2Te_4/(Bi_2Te_3)_n$ to realize different 3D topological phases, and illustrate that a well-localized, half-quantized surface AHC is the signature of the axion insulator phase. Recall that a $MnBi_2Te_4$ monolayer can be effectively described by a $Bi_2Te_3$ monolayer under a FM exchange field[27], we consider a 3D-layered structure composed by vertically stacking 2D TIs, i.e., bilayer $Bi_2Te_3$, with variable separation between bilayers ($d$) and magnetization ($M$). Thanks to the successful synthesis of single-crystal Mn–Bi–Te family and molecular beam epitaxy technique, such multilayer heterostructure could be realized by intercalating an atomic or VdW buffer layer, e.g., BN or $In_2Se_3$, into $MnBi_2Te_4/(Bi_2Te_3)_n$. The model Hamiltonian is written as

$$H = \hbar v_f \tau_z (\boldsymbol{\sigma} \times \mathbf{k})_z + m_\mathbf{k} \tau_x + M \sigma_z$$
$$+ t_{IB}^0 (v_+ \tau_- + v_- \tau_+) + t_{IB}(v_+ \tau_+ e^{ik_z D} + v_- \tau_- e^{-ik_z D}), \quad (1)$$

where $\mathbf{k} = (k_x, k_y, k_z) = (k_\parallel, k_z)$ is the momentum, $\sigma$, $\tau$ and $v$ are Pauli matrices acting on spin, surface, and layer, respectively, with $s_\pm = (s_x \pm is_y)/2$ ($s = \tau, v$; here "surface" refers to the two surface states of each TI monolayer, which are coupled together by the hybridization term $m_\mathbf{k} \tau_x$). The first two terms describe a monolayer $Bi_2Te_3$ with Fermi velocity $v_f$ and Dirac mass $m_\mathbf{k} = (\Delta - Bk_\parallel^2)$[36,37], while the third term denotes the FM exchange coupling between the magnetization $M$ and electrons' spin. The last two terms describe the intra-bilayer and inter-bilayer tunneling, respectively, with $t_{IB}^0$ the intra-bilayer hopping integral and $D$ the superlattice period. For the inter-bilayer hopping, we introduce an exponentially decaying scaling, i.e., $t_{IB} = t_{IB}^0 \cdot e^{-\alpha(d-d_0)/d_0}$, where $d_0$ and $d$ are the intra- and inter-bilayer spacing. The inversion symmetry $I = \tau_x v_x$ is preserved in this Hamiltonian. The model parameters and the analytical solutions of Eq. (1) are provided in Supplementary Note 1.

As shown in Fig. 1, the origin of the phase diagram represents a 3D $Bi_2Te_3$ TI phase. Varying the inter-bilayer coupling with preserved $T$ leads to a $\mathbb{Z}_2$ classification, denoted by the horizontal axis. When the inter-bilayer coupling is weakened, the phase transition between strong TI ($\mathbb{Z}_2 = 1$) and weak TI ($\mathbb{Z}_2 = 0$) occurs. With increasing magnetization, the two TI phases evolve to axion insulator and 3D QSH insulator, respectively, the latter of which can be considered as a trivial stacking of $T$-broken QSH insulator[38]. When the exchange field is strong enough, a 3D Chern insulator phase with chiral side surface states emerges, which is adiabatically connected to a vertical stacking of 2D Chern insulators[39–41]. Under a finite exchange field, the

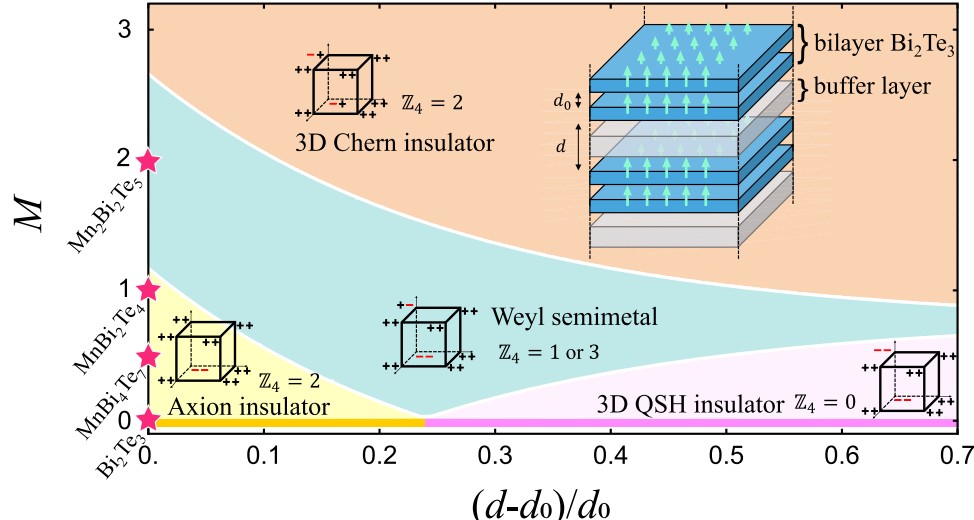

**Fig. 1 Topological phase diagram.** Phase diagram of the multilayer topological heterostructure in terms of relative spacing $(d - d_0)/d_0$ and magnetization $M$ (rescaled as the number of Mn layers per VdW layer) derived from Eq. (1). Insets show the sketch of the heterostructure and $\mathbb{Z}_4$ indices with parities. Four pristine topological materials in FM phase $Bi_2Te_3$ ($M = 0$), $MnBi_4Te_7$ ($M = 1/2$), $MnBi_2Te_4$ ($M = 1$), and $Mn_2Bi_2Te_5$ ($M = 2$) are mapped at the vertical axis.

transition between these three topologically distinct insulating phases inevitably passes an intermediate region, i.e., the Weyl semimetal phase[42].

Depending on the magnetization per VdW layer, the different pristine FM $MnBi_2Te_4/(Bi_2Te_3)_n$ compounds can be mapped onto the vertical axis of the phase diagram with $d = d_0$, as marked by the red stars in Fig. 1. The mapping follows two assumptions: (i) the exchange field in different $MnBi_2Te_4/(Bi_2Te_3)_n$ materials is homogeneous; such assumption works well previously for the mapping of $MnBi_2Te_4/(Bi_2Te_3)_n$ slabs to the 2D topological phase diagram[27]. (ii) The band inversion only occurs at $\Gamma = (0, 0, 0)$ and $Z = (0, 0, \pi/D)$, which means that the band orders at the other inversion-invariant momenta remain the same as that of the nonmagnetic 2D limit ($M = 0, D \rightarrow \infty$), i.e., bilayer $Bi_2Te_3$. Since the Hamiltonian in Eq. (1) has $I$, one can compute the symmetry indicator of $I$, a $\mathbb{Z}_4$ invariant, to determine their topological nature[43–45]:

$$\mathbb{Z}_4 = \sum_{\mathbf{k}=1}^{8} \frac{n_{\mathbf{k}}^+ - n_{\mathbf{k}}^-}{2} \bmod 4 = \sum_{\mathbf{k}=\Gamma,Z} n_{\mathbf{k}}^- \bmod 4, \quad (2)$$

where $n_{\mathbf{k}}^+/n_{\mathbf{k}}^-$ is the number of occupied states with even/odd parity at one of the eight inversion-invariant momenta $\mathbf{k}$. By adding a small magnetization, each doubly degenerate band of the weak TI and strong TI phases splits into two bands with the same parity, leading to $\mathbb{Z}_4 = 0$ and $\mathbb{Z}_4 = 2$, respectively. The former, i.e., 3D QSH insulator, is equivalent to the vertical stacking of 2D $T$-broken QSH insulators with the parities shown in Fig. 1. Such a phase cannot be described by symmetric Wannier functions, while the Wannier obstruction can be removed upon adding a set of trivial elementary band representations (see Supplementary Note 2 for details). Therefore, it corresponds to a distinct type of "fragile topology", manifesting a novel twisted bulk-boundary correspondence[46,47]. The latter corresponds to an axion-insulator phase, as predicted by previous studies using a single $\mathbb{Z}_4$ invariant[18,19,28]. Nevertheless, we find that the 3D Chern insulator phase also yields $\mathbb{Z}_4 = 2$ but a different parity distribution compared with the axion insulator. Therefore, the full indicator group of inversion symmetry $\mathbb{Z}_4 \times \mathbb{Z}_2 \times \mathbb{Z}_2 \times \mathbb{Z}_2$ is required to further distinguish these two phases[48], where the $\mathbb{Z}_2$ indicators

can be chosen as the parity of the Chern number in the $k_i = \pi$ ($i = x, y, z$) plane. As shown in Fig. 1, the full symmetry indicator of the axion insulator phase and the 3D Chern insulator phase are 2:(000) and 2:(111), respectively.

In the phase diagram derived from Eq. (1), $T$ symmetry is always broken, except for the horizontal line with $M = 0$. We find that the three magnetic insulating phases are isolated from each other by an intermediate Weyl-semimetal phase, which is also consistent with the parity analysis. Our symmetry analysis based on DFT calculations shows that all the pristine FM $MnBi_2Te_4/(Bi_2Te_3)_n$ ($n = 0–3$) compounds are $I$-preserved axion insulators. For comparison, we also calculate $Mn_2Bi_2Te_5$ where the magnetic moments per VdW layer are twice as that in $MnBi_2Te_4$, and obtain an unambiguous Weyl semimetal phase, as marked in Fig. 1.

## DFT-calculated surface AHC of magnetic topological phases in $MnBi_2Te_4/(Bi_2Te_3)_n$

Having established the phase diagram, we next calculate the profile of the local Chern marker[49] of the topological phases with nontrivial $\mathbb{Z}_4$ numbers. In principle, one can define a local AHC $\sigma_{AHC} = C_z \cdot e^2/h$, with $C_z$ being the real-space projected Chern number in the $z$-direction. Using the Wannier-representation tight-binding Hamiltonians obtained by DFT-calculated Bloch eigenstates, we compute the local Chern marker $C_z(l)$ projected onto each VdW layer[50,51], expressed as

$$C_z(l) = \frac{-4\pi}{A} \text{Im} \frac{1}{N_k} \sum_k \sum_{vv'c} X_{vck} Y_{v'ck}^\dagger \rho_{vv'k}(l), \quad (3)$$

where $X$ and $Y$ are the position operators along the $x$ and $y$ directions, respectively. $\rho_{vv'}(l)$ is the projection matrix on to the corresponding layer $l$, which implies a summation over all atoms within a VdW layer (see Methods).

The results for the pristine FM $MnBi_2Te_4$ slabs are shown in Fig. 2a. We find that the integrated layer-projected Chern marker $\mathbb{C}(l) = \sum_l C_z(l)$ is stabilized at 1/2 for $l > 2$, and rises up to 1 when it passes over the last two layers, giving rise to a Chern insulator as a whole with $\mathbb{C} = 1$. In this sense, the penetration depth of the surface AHC is about two SLs, while the internal layers do not contribute to the AHC due to the homogeneous spin alignment.

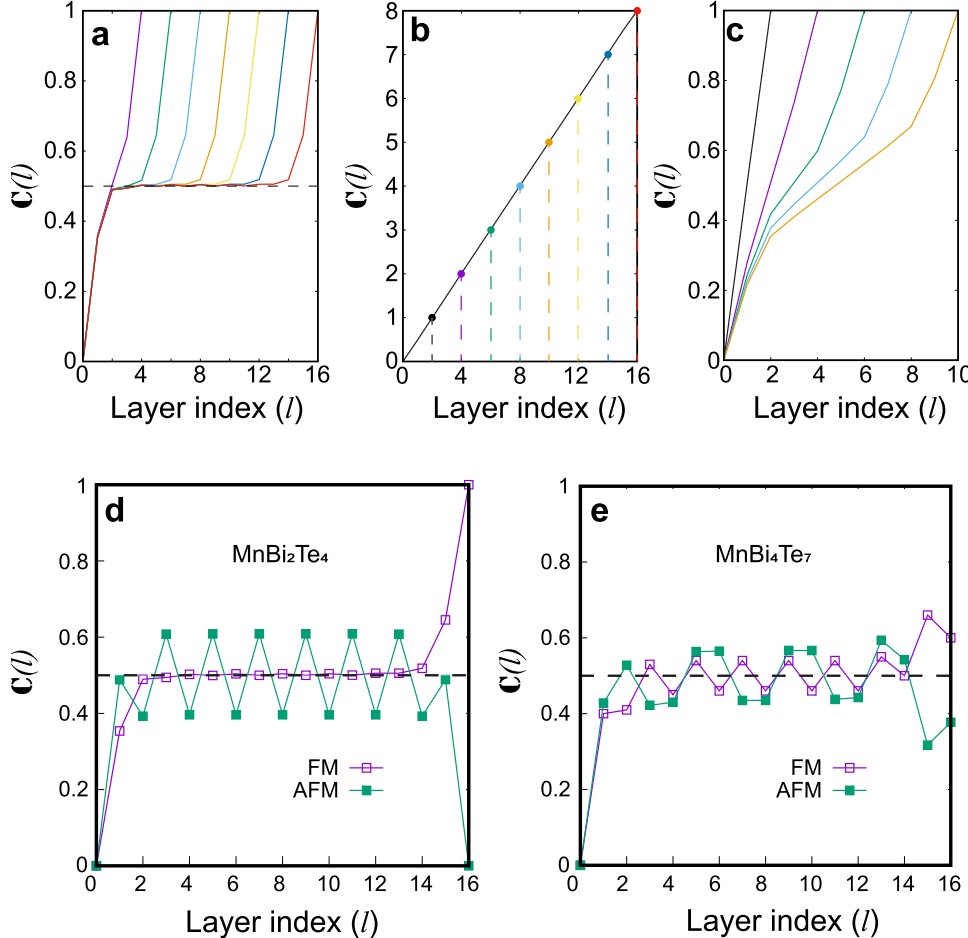

**Fig. 2 Local Chern marker of various topological phases. a–c** Integrated local Chern marker $\mathbb{C}(l)$ for MnBi$_2$Te$_4$ slabs in **a** axion insulator, **b** 3D Chern insulator, and **c** Weyl semimetal phases. In panel **a–c**, the color lines terminate at different positions, denoting the total thickness of the calculated slabs up to 16 VdW layers. **d**, **e** Integrated local Chern marker $\mathbb{C}(l)$ as a function of layer index $l$ for a 16-layer slab of **d** MnBi$_2$Te$_4$ and **e** MnBi$_4$Te$_7$.

Such behavior is similar to the prediction of the axion insulator phase in a nonmagnetic bulk TI with a gapped surface[7]. In addition, the slabs with more than 4 SLs are thick enough to reveal the half-quantized AHC localized at both surfaces, indicating an $I$-preserved, FM axion insulator phase.

The FM MnBi$_2$Te$_4$ turns into a 3D Chern insulator when $d \geq 1.7d_0$, for which the Chern number of a 2D slice within the $k_x$–$k_y$ plane at an arbitrary $k_z$ equals to 1 (Supplementary Note 3). Despite sharing the same $\mathbb{Z}_4$ invariant with the axion insulator, the integrated local Chern number $\mathbb{C}(l)$ of the 3D Chern insulator behaves quite differently. As shown in Fig. 2b, each bilayer contributes to an exact quantized Chern number 1, giving rise to a $\mathbb{C}(l)$ proportional to $l$. Therefore, there is no well-defined surface AHC, instead, the total AHC of the slab is proportional to the total number of primitive cells in the slab.

We note in Fig. 1 that bulk FM MnBi$_2$Te$_4$ falls in the vicinity of the boundary between axion insulator and Weyl semimetal[42], which is thus sensitive to numerical details such as the choice of exchange-correlation functionals and lattice constants[24,26]. Figure 2c shows the corresponding $\mathbb{C}(l)$ for the Weyl semimetal phase obtained by applying a 1% lattice expansion. It is found that for the insulating slabs (Supplementary Note 3), the surface AHC is no longer quantized to 1/2 due to the bulk contribution. Especially for the slabs thicker than five VdW layers, the surface AHC contribution from the top and bottom two layers ranges

from 0.42 to 0.35, while the internal-layer contribution increases linearly with the number of layers due to the bulk AHC[52]. For all these slabs, the distribution of the surface AHC is not localized at one surface, but extends to the center of the slab.

## Half-quantized surface AHC in the axion insulator phases

We now focus on the axion insulator phase and its surface AHC in FM and AFM MnBi$_2$Te$_4$/(Bi$_2$Te$_3$)$_n$ compounds. The magnetic ground states for MnBi$_2$Te$_4$, MnBi$_4$Te$_7$, and MnBi$_6$Te$_{10}$ ($n = 0$–2) are A-type AFM along the $z$-axis with the local moments of Mn FM ordered within each MnBi$_2$Te$_4$ layer[17,53], while MnBi$_8$Te$_{13}$ ($n = 3$) is a ferromagnet[18,21]. From the perspective of $\mathbb{Z}_4$ indices, all of the above-mentioned compounds, no matter FM or AFM states, are $I$-preserved axion insulators according to our DFT calculations. For slab calculations, the total Chern number depends on whether $I$ is preserved in the slab geometry. Therefore, for FM and odd-layer AFM MnBi$_2$Te$_4$ slabs with $I$, the total Chern number reaches to 1 because both top and bottom surfaces contribute the same AHC $e^2/2h$. On the other hand, as shown in Fig. 2d, the topmost layer of the even-layer AFM phase (with broken $I$) contributes almost half-quantized AHC, i.e., $C_z(1) = 0.49$, while the bottom layer contributes an opposite AHC $C_z(16) = -0.49$, leading to a zero Hall plateau state and a zero total Chern number. Starting from the second layer, $\mathbb{C}(l)$ no longer saturates but oscillates around 1/2 with a period of the unit

cell in the $z$-direction, i.e., two VdW layers. Every additional layer contributes reversely to $\sigma_{AHC}$ due to the flipping spin direction, leading to the oscillation with the amplitude as large as $0.21e^2/h$. Such behavior is in sharp contrast to $T$-preserved axion insulators proposed before. For the case of FM and AFM MnBi$_4$Te$_7$, the layer-projected AHC at MnBi$_2$Te$_4$ termination reaches the half-quantized value after three or four VdW layers, then oscillates around $e^2/2h$, again, due to the inhomogeneity of the magnetic moments. The oscillation period is also determined by the thickness of a unit cell, i.e., 2 (4) VdW layers for FM (AFM) phase. The details are provided in Supplementary Note 4.

It is worthwhile to note that the half-quantized AHC of an axion insulator is a local property at the gapped surface[54]. To demonstrate this, we consider a thick slab of FM MnBi$_4$Te$_7$, which is insulating for the MnBi$_2$Te$_4$ termination but metallic for the Bi$_2$Te$_3$ termination. We find that as long as the Fermi level ($E_f$) locates within the surface gap of the MnBi$_2$Te$_4$ termination, the corresponding surface AHC would stay around $e^2/2h$. On the other hand, the surface AHC with the metallic Bi$_2$Te$_3$ termination varies with different choices of $E_f$ (see Supplementary Note 4). Overall, the locality of the surface AHC does not rely on the metallicity of the whole slab, but is due to the vanishing contribution of the local Chern marker from the bulk state, i.e., $\sum_{l\in internal}^{l+u.c.}C(l)=0$. This can also be used as a criterion to distinguish the "surface" and "bulk" layers.

## Spectral signatures of the surface AHC—hinge states

It is of great importance to consider how the computed local Chern marker and surface AHC corresponds to a measurable physical quantity[2], thus providing direct evidence for the axion-insulator phase. To be specific, one might wonder what kind of "mode" carries the half-quantized AHC in a realistic material. Unlike the 2D system with the well-defined 1D edge, 2D surfaces of a 3D material are terminated by hinges between different surfaces. While the top and bottom surfaces of MnBi$_2$Te$_4$/(Bi$_2$Te$_3$)$_n$ are both gapped by the out-of-plane magnetization, the side surfaces are either gapless or gapped depending on the AFM or FM configuration, respectively. Combining DFT calculations and recursive Green's function approaches (see Methods), we calculate the real-space local density of states (LDOS) of the

surface and hinge states of a semi-infinite sample, as shown in Fig. 3. Fortunately, the hinge states of both AFM and FM axion-insulator phases provide effective signatures that could be detected by experiments. We next discuss the two cases separately.

For the side surface of AFM MnBi$_2$Te$_4$, a gapless Dirac cone occurs due to the combined symmetry $T\tau_{1/2}$, where $\tau_{1/2}$ denotes the half-cell translation along the stacking axis[55]. Similar to the nonmagnetic TI, the manifestation of the bulk magnetoelectric response at the side surface is compensated by the opposite contribution from the surface Dirac cone, leading to vanishing side-surface AHC. Instead, there exists helical modes with the opposite spin channels propagating through opposite directions. The terminated AFM MnBi$_2$Te$_4$ sample and the calculated LDOS of the top hinge and side surface states are shown in Fig. 3a–c (position ② and ③). Compared with the helical gapless side surface states, we find that a remarkable feature of the hinge state is the asymmetric spectral weight between the left- and right-moving modes, indicating its chiral nature. We denote such chiral hinge modes embedded inside the side surface bands as "in-band hinge" states, and attribute them to the difference of the surface AHCs of the top and the side surfaces[31] (see Supplementary Note 6 for details). For even-layer AFM slabs where the top and bottom surfaces have opposite magnetizations, the top and bottom in-band hinge states manifest opposite chiralities accordingly.

For FM MnBi$_2$Te$_4$, the side surface is gapped by the $z$-direction magnetization and a lower crystal symmetry, stemming from a high $k$-order effect of spin-momentum locking. In such a situation, the in-band hinge states still exist, with the top and bottom hinges having the same chirality, as shown in Figs. 3e and 3f. In addition, such a side surface exhibits a half-quantized local Chern marker, leading to a single chiral mode (½ + ½) at the top hinge and no chiral modes (½ − ½) at the bottom hinge, inside the side surface gap (position ② and ④, see Fig. 3f). Such a chiral hinge mode with integer AHC denoted as in-gap hinge mode, is equivalent to the chiral domain wall state in high-order topological insulators[56,57], which is also predicted in FM MnBi$_2$Te$_4$/(Bi$_2$Te$_3$)$_n$ by model calculations[28]. However, we note that such in-gap hinge mode only exists within a small energy range, i.e., 6 meV for FM MnBi$_2$Te$_4$ because its side surface gap is a high-

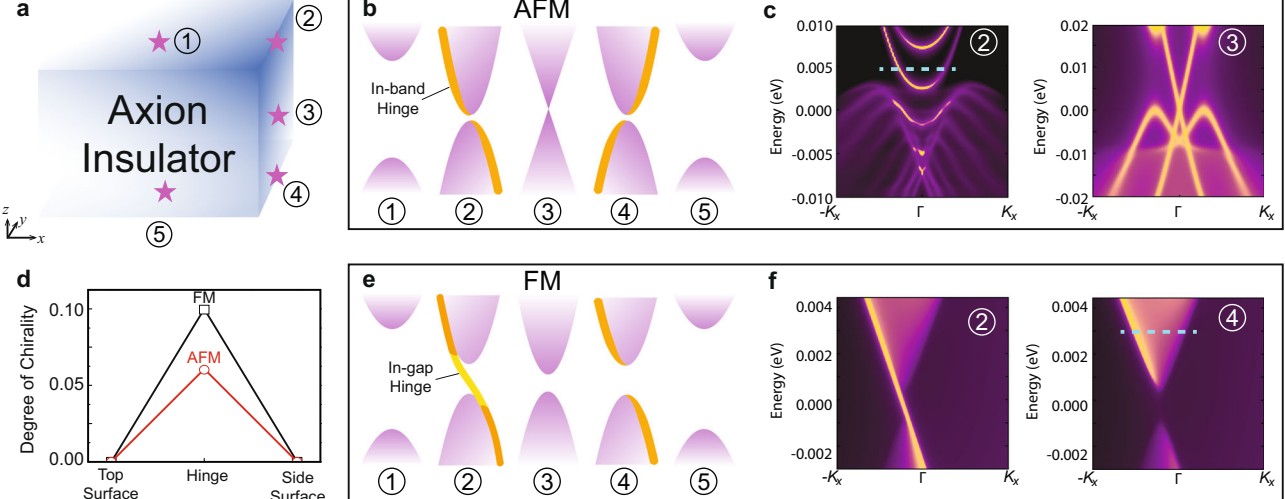

**Fig. 3 Spectra of surface and hinge states. a** Schematic of an axion insulator, with its top surface, top hinge, side surface, bottom hinge, and bottom surface marked as ①-⑤, respectively. **b,e** Schematic of the topological band spectra at spots ①-⑤ for **b** AFM and **e** FM MnBi$_2$Te$_4$. The orange and yellow lines denote in-band hinge and in-gap hinge states, respectively. **c, f** The hinge and the side surface LDOS of **c** AFM MnBi$_2$Te$_4$ (② and ③) and **f** FM MnBi$_2$Te$_4$ (② and ④). **d** The degree of chirality of the in-band hinge states with the Fermi level marked by the blue dashed lines in **c** and **f**.

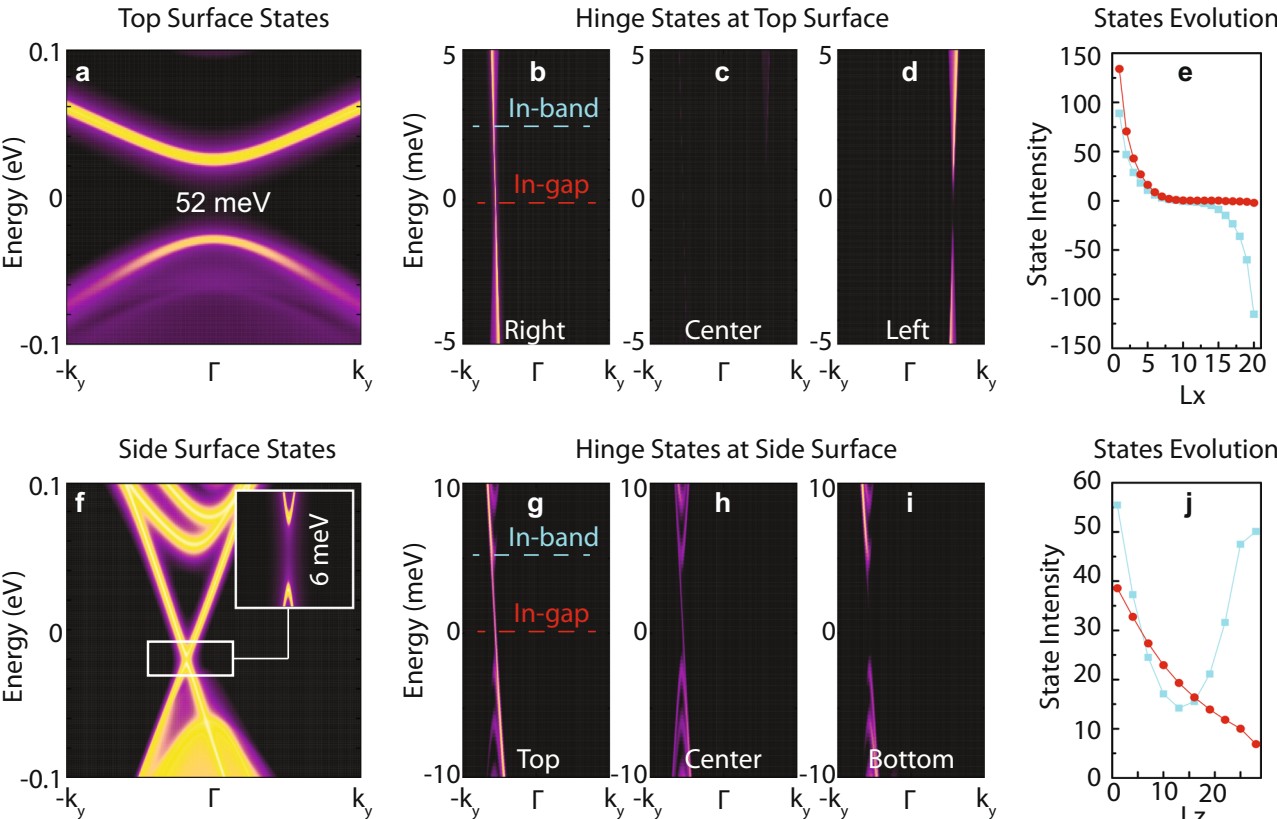

**Fig. 4 Real-space evolution of the hinge spectra.** Surface states and the evolution of the hinge states through the top surface **a**–**e** and the side surface **f**–**j** for FM axion insulator $MnBi_2Te_4$. **a** Top surface states. **b**–**d** LDOS of a sample with 20-unit-cell thick in the $x$-direction, including the right hinge **b**, center of the top surface **c**, and the left hinge **d**. **e** Intensity of the in-band (blue) and in-gap (red) hinge states projected onto different positions of the top surface as a function of the unit-cell index (#1 and #20 are the hinges, while #10 is the center of the computed cell). **f** Side surface states. **g**–**i** LDOS of a sample with 30-unit-cell thick in the $z$-direction (i.e., 30 VdW layers), including the top hinge **g**, center of the side surface **h**, and the bottom hinge **i**. **j** Similar to **e** but for the side surface.

order magnetization gap. On the other hand, the in-band hinge modes for both of the top and bottom hinge, originated from the half-quantized surface AHC, remain robust within a much larger energy range, i.e., the top/bottom surface gap (52 meV), which is favorable for experimental detection.

Besides the non-integer and integer topological charge, another feature to compare the in-band hinge states and the in-gap hinge states is their decay length in the real space, which clearly reflects their distinct topological origins. Figure 4 shows the LDOS of the hinge states projected onto different positions of the top surface (along the $x$-direction, Fig. 4a–e) and the side surface (along the $z$-direction, Fig. 4f–j), respectively. We find that the in-gap hinge state decays rapidly along the horizontal ($x$) direction at the top surface, while it decays much more slowly along the vertical ($z$) direction at the side surface. Specifically, five-unit-cell along the $x$-direction at the top surface is long enough for the in-gap hinge states to drop to below 1/10 of the maximal spectral intensity at the hinge, while at the side surface, the in-gap hinge states only decay to 1/7 of the maximal spectral weight within a thickness of 30 VdW layers along the $z$-direction. This seems counterintuitive due to the weak VdW interaction between layers along the $z$-direction. However, it can be understood by taking into account the surface bandgap: the side surface gap is much smaller than the top surface gap due to the magnetization direction, leading to a much longer wavefunction decay length[58]. On the other hand, the in-band hinge states reverse their chiralities on different sides of the top surface (see Figs. 4b, 4d and 4e), and decay rapidly through both of the top and the side surfaces (blue curves in

Figs. 4e and 4j), which agrees well with the locality of the surface AHC shown in Fig. 2d. Such consistency again demonstrates that the in-band hinge state is an ideal physical quantity to verify the existence of the surface AHC of the axion insulator phase. Note that the in-band hinge profile in Fig. 4j also distinguishes that of a FM trivial insulator and a Chern insulator on top of a trivial insulator.

The in-band hinge states will contribute to unique transport signatures when $E_f$ crosses the side surface bands. As shown in Fig. 3d, compared with nonchiral top and side surface states, these localized chiral in-band hinge states exhibit imbalanced spectral weight for the left-moving and right-moving modes. We note that even for FM axion insulators, one can probe a specific hinge with only in-band hinge contribution (position ④ in Fig. 3f). We thus propose a device setup with a thick-enough sample and multiterminal leads attached to one surface covering only a few VdW layers. While the signal of in-band hinge modes is buried by the metallic side surface states for typical two-terminal transport measurements, one can expect nonzero signal through nonlocal surface transport measurements[59]. Although the exact number of $e^2/2h$ conductance is not topologically protected and not immune from subtle device structure and disorder effects, the chiral in-band hinge states still give rise to unambiguous transport signature as the direct evidence of axion insulators, which is in sharp contrast to the case of a trivial insulator[60]. We also provide a spectral comparison between a FM axion insulator and a FM trivial insulator in Supplementary Note 7.

## Discussion

A gapped surface state is the kernel to realize the axion insulators. While recent ARPES measurements unexpectedly show an almost gapless surface Dirac cone in $MnBi_2Te_4$[60–62], we briefly provide two promising candidates with surface gaps in $MnBi_2Te_4/(Bi_2Te_3)_n$ family in Supplementary Fig. 8. By comparison of our ARPES and DFT results, we can distinguish two different types of surface gaps depending on the specific terminations. Type I originates from the typical surface magnetization that introduces a $M_z\sigma_z$ term to a gapless Dirac fermion, exemplified by the (001) surface of $MnBi_2Te_4$, and the $MnBi_2Te_4$ termination of $MnBi_2Te_4/(Bi_2Te_3)_n$. Although the surface states of $MnBi_2Te_4$ are reported to be gapless due to the possible reconstruction of the geometric or magnetic configurations at the surface[60–62], there are still unambiguous surface gaps observed in various conditions, including Sb-doped $MnBi_2Te_4$ and the $MnBi_2Te_4$ termination of $MnBi_8Te_{13}$[21,63,64]. Type II, on the other hand, is caused by the hybridization effect between the Dirac cone and the bulk valence band, exemplified by the $Bi_2Te_3$ termination of $MnBi_4Te_7$[65]. Compared with the magnetization gap, the hybridization gap exchanges the orbital characters of the surface band and bulk band. Nevertheless, the broken $T$ also gives rise to an imbalanced Berry curvature with opposite momenta and thus a half-quantized AHC, which is also provided in Supplementary Note 8.

To summarize, we demonstrate that the half-quantized surface AHC in $MnBi_2Te_4/(Bi_2Te_3)_n$ series is well localized at a few layers from the top/bottom surfaces, which could be an experimental observable. The surface AHC is represented by the chiral hinge mode embedded inside the side surface bands, providing guidelines for nonlocal transport measurements. Our finding establishes an ideal platform to realize the long-sought axion states and the related topological magnetoelectric phenomena. Besides, the fruitful topological phase diagram, including 3D Chern insulators, Weyl semimetals, and fragile 3D QSH insulators, attributes new possibility to this family in the search of novel quantum materials.

## Methods

**DFT calculations**. DFT calculations are performed using Vienna Ab-initio Simulation Package (VASP)[66] to provide insights on electronic structures in the $MnBi_2Te_4/(Bi_2Te_3)_n$ system. The generalized gradient approximation developed by Perdew, Burke, and Ernzerhof (PBE)[67] is used to describe the exchange-correlation energy in our calculation. The projector augmented wave (PAW) method is used to treat the core and valence electrons using the following electronic configurations: $3p^64s^23d^7$ for Mn, $5d^{10}6s^26p^3$ for Bi, and $5s^25p^4$ for Te. The electron correlation effects of Mn-3d states are considered by the inclusion of the Hubbard U (PBE+U), with $U(Mn) = 5$ eV. The Brillouin zone is sampled by an $8 \times 8 \times 1$ $\Gamma$-centered Monkhorst–Pack $k$-point mesh. Once the electronic structure is converged, the Bloch states are projected to the Wannier functions[68,69] of the Mn-3d, Bi-6p, and Te-5p orbitals to build the tight-binding Hamiltonian.

The calculation of local Chern number for a particular VdW layer ($l$) is systematically derived by Varnava et al. in ref.[51], and expressed as Eq. (3) in the main text, where $A$ is the unit cell area, $X(Y)_{ijk} = \frac{\langle \psi_{ik} | i\hbar v_{x(y)} | \psi_{jk} \rangle}{E_{ik} - E_{jk}}$ is the matrix element for the position operator along the $x$ or $y$ directions, and the band indices go through the conduction ($c$) and valence ($v$) bands as denoted for the summation. $N_k$ is the number of $k$-points and $\rho(l)$ the projection matrix onto the orbitals of the atoms within the corresponding VdW layer.

**Calculation of the hinge states**. We have employed two DFT-based methods to compute the hinge states, as plotted in Fig. 3 and Fig. 4. Both methods show consistent chiral states at the hinge of the $MnBi_2Te_4$ system. In Fig. 3, the hinge states are calculated using a bi-semi-infinite open-boundary geometry condition, which is also used in ref.[70]. The structure is semi-infinite along the $x$- and $z$-directions, while the periodic boundary condition (PBC) is maintained along the $y$-direction so that $k_y$ remains a good quantum number. The Hamiltonian can be written in terms of a quasi-block-tridiagonal form as follows:

$$H = \begin{pmatrix} H_0 & H_1^x & 0 & 0 & \cdots & H_1^z & 0 & \cdots \\ H_1^{x\dagger} & H_0 & H_1^x & 0 & \cdots & & & \\ 0 & H_1^{x\dagger} & H_0 & H_1^x & \ddots & & & \\ 0 & 0 & H_1^{x\dagger} & H_0 & \ddots & & & \\ \vdots & \vdots & \ddots & \ddots & \ddots & & & \\ H_1^{z\dagger} & & & & & H_0 & H_1^z & 0 & \cdots \\ 0 & & & & & H_1^{z\dagger} & H_0 & H_1^z & \ddots \\ \vdots & & & & & 0 & \ddots & \ddots & \ddots \end{pmatrix}, \quad (4)$$

where $H_0, H_1^x$ and $H_1^z$ are the ground-state Hamiltonian and hopping matrices along the $x$- or $z$-directions, respectively. Such a tight-binding Hamiltonian is obtained from the maximally localized Wannier functions constructed by the wannier90 package interfaced to the VASP code. Then the Hamiltonian can be written into

$$H = \begin{pmatrix} H_0 & H_I \\ H_I^\dagger & H_R \end{pmatrix}. \quad (5)$$

One can immediately define a Green's function for such Hamiltonian following the traditional scheme, i.e.,

$$G = \begin{pmatrix} G_0 & G_I \\ G_I^\dagger & G_R \end{pmatrix}. \quad (6)$$

Note that $G_R$ includes the surface Green's function along both the $x$- and $z$-directions, which can be computed iteratively.

In Fig. 4, the evolution of the hinge states is investigated as a function of distance away from the hinge. Then a supercell with finite width along the $x$- (for the top surface) or the $z$-direction (for the side surface) is needed. The system geometries are a) For the top surface, finite along $x$ (20-unit-cell-thick), semi-infinite along $z$, PBC along $y$; b) For the side surface, finite along $z$ (30-unit-cell-thick), semi-infinite along $x$, PBC along $y$. Surface states are computed iteratively as the retarded Green's function along the semi-infinite direction. More details can be found in Supplementary Note 5.

## Data availability

The supportive data for the findings in this study are available from the corresponding authors upon reasonable request.

## Code availability

The computation code for getting the theoretical prediction is available from the corresponding authors upon reasonable request.

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

## Acknowledgements

We thank Ni Ni, Yue Zhao, and Zhongjia Chen for helpful discussions. This work was supported by the National Key R&D Program of China under Grant Nos. 2020YFA0308900 and 2019YFA0704900, the National Natural Science Foundation of China under Grant No. 11874195, Guangdong Innovative and Entrepreneurial Research Team Program under Grant No. 2017ZT07C062, Guangdong Provincial Key Laboratory for Computational Science and Material Design under Grant No. 2019B030301001, the Shenzhen Science and Technology Program (Grant No. KQTD20190929173815000), and Center for Computational Science and Engineering of Southern University of Science and Technology.

## Author contributions

Q.L. conceived the idea and supervised the research with J.-P.L., M.G., H.S., and Y.Z. performed DFT calculations. M.G. and J.-P.L. developed the Wannier orbital-based tight-binding model to compute the layered Chern marker and hinge states. J.-Y.L. constructed the model Hamiltonians and computed the topological phase diagram. H.L. contributed to the proposal of nonlocal transport measurements. C.L. provided the ARPES data for comparison. Q.L. wrote the paper with the input from all authors.

## Competing interests

The authors declare no competing interests.
