## [Peer Review File · Nature Communications]

Reviewers' comments:

Reviewer #1 (Remarks to the Author):

Mingqiang Gu et al. investigate the anomalous Hall effect in different magnetic topological phase such as axion, magnetic Weyl, and 3D Chern insulator, by using k.p model. In addition, they take $\text{MnBi}_2\text{nTe}_{3\text{n}+1}$ and $\text{Mn}_2\text{Bi}_2\text{Te}_5$ as the examples to support the results of their k.p model. Since 3D Chern insulator has not been discovered yet in real materials, authors try to realize this phase via tuning the hopping parameters in MnBi_2Te_4 . In the second part, authors report the real space distribution of anomalous Hall conductivity, especially focus on axion and Weyl phase. Finally, the authors show the real space surface and hinge states calculations of MnBi_2Te_4 to provide the possibility of experimental measurement of $0.5 e^2/h$.

(1) In this article, the author sometimes calls the FM $\text{MnBi}_2\text{nTe}_{3\text{n}+1}$ axion and sometimes the AFM $\text{MnBi}_2\text{nTe}_{3\text{n}+1}$ is axion, which makes the article difficult to read. It is recommended that the authors explain this part clearly. In addition, what is the difference between the axion state of $\text{MnBi}_2\text{nTe}_{3\text{n}+1}$ and the AFM sandwich structure FM(up)/TI/FM(down)?

(2) In the k.p model, authors assume the FM exchange field in all layers are the same, and adjust the parameter "M" to present different $\text{MnBi}_2\text{nTe}_{3\text{n}+1}$ ($n=1\sim 2$). For example, MnBi_4Te_7 ($M=1/2$) and MnBi_2Te_4 ($M=1$). In real material of MnBi_4Te_7 , there are two different species of unit blocks, MnBi_2Te_4 (septuple layer, SL) and Bi_2Te_3 (quintuple layer, QL). I am wondering why the Bi_2Te_3 cannot be regarded as a buffer layer in the k.p model, but as having the same exchange field as MnBi_2Te_4 .

(3) I am wondering why the homogeneous spin alignment do not contribute to the AHC, while inhomogeneous spin will provide the AHC. In addition, in Fig. 2(d) and (e), I am wondering why profile of MnBi_2Te_4 and MnBi_4Te_7 are quite different. For example, in FM phase, the AHC in MnBi_2Te_4 shows a 0.5 plateau and reach $C=1$ as $l=16$, while MnBi_4Te_7 shows an oscillation and does not increase significantly as $l=16$.

(4) In the 2nd paragraph on p.6, authors mention the FM MnBi_2Te_4 turns into 3D Chern insulator as reducing the hopping parameter between the MnBi_2Te_4 bilayer. In this phase, the Chern number of a 2D slice within the k_x - k_y plane at an arbitrary k_z equals to 1. Authors also mention, in this phase, each bilayer contributes to an exact quantized Chern number 1. This result implies that 3D Chern insulator can be view as 2D Chern insulator stacking along z direction. However, monolayer MnBi_2Te_4 is trivial insulator, not 2D Chern insulator. I am wondering if author can explain more clearly in this part.

(5) In the p.7, line 198-200, "Firstly, while the axion insulator phase cannot be fully characterized by $\mathbb{Z}_4=2$, it manifests a surface AHC saturated at or oscillates around $e^2/2h$." I am confused by this sentence, because in p.3, authors use \mathbb{Z}_4 to identify axion phase.

(6) In the 2nd paragraph on p.9, authors propose that there is a hinge state on the top surface but there is no hinge state on the bottom surface. They mention the domain wall of this system as a higher-order TI. I am confused by this paragraph. In HOTI, the hinge state will go around the edge of 3D system obey the crystal symmetry. It means, if three directions of system are finite, we start from the top surface and moving along the hinge state, we will reach the bottom surface via the side edge, and finally return to the upper surface, complete a 1D closed loop. I am wondering if author can explain more clearly in this part.

(7) Fig.2, the caption does not specify the number of layers represented by different color lines. (b) and (c) are incorrectly marked.

(8) Fig.3(a) is the side surface state of AFM MnBi₂Te₄. Previous work reports the ground state of AFM MnBi₂Te₄ is an AFM-TI. I am wondering why the Dirac point in Fig.3(a) is not on the Gamma point. The result of Fig.3(a) seems like a topological crystalline insulator.

Reviewer #2 (Remarks to the Author):

In this manuscript, the authors provided a systematic study on the surface anomalous Hall conductance (AHC) for the recently discovered topological magnetic materials Mn(Bi,Te). Based on the first-principles-based calculations of local Chern marker, they managed to extract the AHC for each layer in a finite-size slab of various magnetic topological phases, and summarized several interesting features for the AHC. I think the current paper contains some interesting results that might be publishable. However, some results and statements seem quite confusing and likely NOT valid. Meanwhile, I find the current draft hard to read and the authors should spend lots of efforts to improve the readability, including fixing the typos and refining logical flow in writing. In addition, I am not convinced that the novelty level of this work satisfies the high standards of Nature Comm., given that some results might have already existed in the literature. Therefore, I cannot recommend this work for publication in Nature Comm.

Here're some comments to support my judgement:

1. While the authors provide a detailed ab-initio-based studies on the surface AHC of a specific family of materials, the main conclusion they have reached, which I believe is the surface localization of AHC, do not surprise me at all. For example, it has been a common sense in the community that axion insulators generally host half-quantized surface AHC, which, by definition, should be localized around the surface. This surface localization of AHC can be intuitively understood by considering a magnetic domain wall on the surface of a 3D TI, which is known to host a chiral domain wall mode. Since such chiral mode signals the sign change of half-quantized surface AHC, its localization around the surface should also implies the localization of surface AHC.

2. It seems to me that authors failed to acknowledge some contributions in the previous literature, which is somewhat disappointing. In particular, as far as I understand, part of the results in this work seems to have some overlaps with the model study in Ref. 27, which, however, is only mentioned very briefly in the introduction. Below, I list a few examples for the authors to think about:

(i) The Z₄ indicator for ferromagnetic axion insulating phase in Mn(Bi,Te) was simultaneously and independently calculated in the following references A & B (see below) as well as in Ref. 27.

(ii) Can the authors explain the difference between the effective Hamiltonian in Ref. 27 and the one in Eq. 1 for this work? I find these two Hamiltonians quite similar to each other, and I think the authors should address the similarities and/or discrepancies in the main text, when Eq. 1 first shows up.

(iii) As far as I am aware of, Ref. 27 appears to be the first prediction of chiral hinge modes in FM Mn(Bi, Te). Even though the hinge mode calculation is regarded as an important result of the current paper, the contribution of Ref. 27 is not mentioned at all. Can the authors explain the similarities and differences between their hinge mode calculations with those in Ref. 27?

(iv) In section "Surface AHC of magnetic topological phases", the authors mentioned that the FM axion insulating phase is found to have a saturation of Chern number quantized to one, as the sample thickness increases. This is exactly the higher-order TI phase proposed in Ref. 27, am I correct?

[A] Hu, Chaowei, et al. "Realization of an intrinsic ferromagnetic topological state in MnBi₈Te₁₃."

Science advances 6.30 (2020): eaba4275.

[B] Tian, S., Gao, S., Nie, S., Qian, Y., Gong, C., Fu, Y. & Shin, S. (2020). Magnetic topological insulator MnBi₆Te₁₀ with a zero-field ferromagnetic state and gapped Dirac surface states. Physical Review B, 102(3), 035144.

3. The authors defined a 3D fragile QSH insulator and claimed it to have fragile topology without providing any supporting evidence. To me, the concept of fragile topology has a rigorous definition. Namely, a fragile topological insulator by itself cannot be described by symmetric Wannier functions, while such a Wannier obstruction is "fragile" and can thus be removed upon coupling to a set of trivial atomic bands. Therefore, the authors will need to prove the fragile topological nature based on the above definition before making such claims.

4. I find the last paragraph of the section "Experimental realization of surface AHC" rather confusing. The authors conclude that for the FM axion insulating state, the chiral hinge mode only circulates around the top surface, but not the bottom surface. This does not make sense to me, since the FM state does have a spatial inversion symmetry (based on which the Z₄ indicator is calculated). In fact, the absence of hinge mode for the bottom hinge can be naturally explained by Fig. 4c of Ref. 27, even considering inversion symmetry. As shown in Fig. 4c of Ref. 27, both C₃ and inversion symmetries enforce a special pattern of the chiral hinge mode around the sample, such that for a given side surface, the hinge mode is always absent on either the top or the bottom hinge. In this regard, I think the authors have falsely assumed that a uniform magnetic domain wall is formed between the top surface and every side surface, as they discussed in Supplementary Note 5. In fact, as pointed by Ref. 27 in the appendix, it is the hexagonal warping term in Mn(Bi, Te) that makes the domain wall to appear only in alternating side surfaces, which causes the hinge mode pattern in Fig. 4c.

To clarify the hinge mode pattern, I would suggest the authors to calculate not only the top & bottom hinge modes on the same side surface, but also those on a parallel side surface on the back of the sample. I believe for the other side surface, the hinge mode can only appear on the bottom hinge, according to the Fig. 4c of Ref. 27.

If the hinge mode pattern is indeed the one in Fig. 4c of Ref. 27, does the proposed non-local transport setup still work?

5. While reading the paper, I am often confused about which of the results are obtained via first-principles calculations and which are from the effective Hamiltonian. In addition, most discussions have involved both AFM phase and FM phase. Sometimes, I could easily get lost while reading a specific statement, being not clear about which phase the statement is referred to. I would suggest the authors to improve the writing of the draft and makes it easier for readers to follow.

6. I don't quite understand the physical origin of the chiral hinge mode that coexists with the gapless surface state in the AFM phase. Can the authors elaborate on this?

7. In the last paragraph of section "Multiple topological phases...", the authors mentioned "any transition between two of the three insulating phases must be connected by an intermediate Weyl-semimetal phase". This is a very strong statement. I think it is generally possible (maybe in a different model) that the phases are separated by a line of topological phase transition, but not a finite region.

8. Can the authors provide the model parameters for Eq. 1?

9. I think there is a typo in the last two paragraphs of the section "Surface AHC of magnetic topological phases", where Fig. 2b and Fig. 2c are messed up with each other.

Reviewer #3 (Remarks to the Author):

In this manuscript, the authors studied the Hall effect and the quantization of the Hall conductance on the axion insulator's surface. A phenomenological model was constructed as stacked 2D TI layers with interlayer couplings and a ferromagnetic Zeeman field. This model has a rich phase diagram, including the axion insulator, the Weyl semimetal, the 3D Chern insulator, and a magnetic fragile topological insulator. This model well explains the various topological phases in the popular material family $\text{MnBi}_2\text{Te}_4/(\text{Bi}_2\text{Te}_3)_n$. The authors also applied DFT calculation to the materials $\text{MnBi}_2\text{Te}_4/(\text{Bi}_2\text{Te}_3)_n$ and computed the local Chern markers, from which they found that the top or bottom surface of an axion insulator is either $1/2$ or $-1/2$. The $+1/2$ Chern markers are consistent with the theoretical prediction of half-quantized Hall conductance of a single surface of the axion insulator.

The manuscript is well written, and the calculations are solid. In light of the high standards of Nature Communication, I think a key aspect in the assessment of the present manuscript should be a clear prediction of the unique transport properties in the experiment. In my opinion, however, the proposed experiments are not well explained. Therefore, I can recommend publishing in Nature Communication if the authors could address the following comments satisfactorily in a revised version.

(1) In the section "Half-quantized surface AHC in axion insulator phases" and in Figure 2, it is shown that the FM axion insulator has a total Chern number 1 whereas the AFM axion insulator has a total Chern number 0. My question is whether this statement depends on the slab's configuration, e.g., the even/odd layers' effect. From the aspect of inversion symmetry, which stabilizes the axion phase, there is no crucial difference between the FM and AFM axion insulators. In general, I believe that the total Chern number of an axion insulator must be ± 1 if the sample respects the inversion symmetry. From Supplementary Figure 2, the AFM slab does break the inversion symmetry. I wonder whether an additional bottom layer with spin pointing up will recover the inversion symmetry and make the total Chern number odd.

(2) The local Chern markers do show half-quantization around the surfaces. However, it is known that it is not clear how this half-quantization is reflected in the experiment. In the section "Experimental realization of surface AHC", the authors discussed two different cases: (a) the side surface is almost gapless (b) the side surface is gapped. Case (a) is not well explained. I think the sentence "One is the finite-size gap originated from the coupling between the top and bottom surfaces" should be "One is the finite-size gap originated from the coupling between the left (front) and right (back) surfaces" because the top and bottom surfaces are assumed as gapped in the first sentence in this paragraph. The finite size gap is exponentially small in a macroscopic or mesoscopic sample. So I am not sure whether the discussion based on the finite-size gap is relevant. Also, the statement "nonlocal transport signature of the chiral hinge state appears when the Fermi level crosses the side surface bands ..." is also confusing: what's the configuration of the chiral hinge states in real space? What kind of nonlocal transport signature? Why does the Fermi level need to cross the surface bands?

(3) In case (b), the side surface is gapped and contributes to an additional $1/2$ or $-1/2$ AHC's to the top and bottom surfaces, leading to integer AHC on the top (1) and bottom (0) surfaces. My question is how to distinguish this top surface with AHC 1 from a 2D QAH on top of a trivial insulator? The two situations have the same surface AHC but different bulk topology.

(4) The authors also mentioned that "Such measurements serve as the direct evidence of axion insulators, which is in sharp contrast to the case of a trivial insulator". For the paper to be more self-

contained, it is better to explain such measurements and explicitly point out the sharp difference.

The followings are some minor comments.

(5) It would be good if the authors can clarify the symmetries that equation 1 has. For example, the inversion operator and the combined operation of time-reversal and translation in the AFM phase.

(6) Equation 1 does not have a Brillouin zone; hence equation 2 can not be directly applied. The authors must assume the parities at $(k_x, k_y) = (\pi, 0), (0, \pi), (\pi, \pi)$ for $k_z = 0$ and π . It is helpful to explain how this equation is evaluated.

(7) The 3D Chern insulator indeed has the same z_4 index as the axion insulator. However, this does not mean that the 3D Chern insulator cannot be distinguished from the axion insulator through the symmetry-based indicators. Considering the inversion symmetry, the indicator group is $Z_4 * Z_2 * Z_2 * Z_2$, where the Z_2 indicators can be chosen as the parity of the Chern number in the $k_i = \pi$ ($i = x, y, z$). It is known that when $z_2 = 1$, which is true in the 3D Chern insulator, $z_4 = 2$ no longer corresponds to the axion insulator.

Summary of the main changes:

1. To clarify the most important contribution of our work, a number of changes are made. We have performed more calculations and analysis on the hinge states to elucidate the spectral signatures of the surface anomalous Hall effect in $\text{MnBi}_2\text{Te}_4/(\text{Bi}_2\text{Te}_3)_n$ axion insulator phases. Accordingly, the relevant section is completely rewritten (from Page 8 to 10) and renamed as “*Spectral signatures of the surface AHC – hinge states*”, with a revised Fig. 3 and a new Fig. 4. We have employed two DFT-based methods to compute the hinge states, with the methods updated in the Methods section.
2. To highlight the main contribution of our work, we have changed the title to “*Spectral signatures of the surface anomalous Hall effect in magnetic axion insulators*”.
3. We also complement the abstract to emphasize our main achievement. It reads: “*We also discuss the resultant chiral hinge modes embedded inside the side surface bands as the potential experimental signatures for transport measurements.*” in Page 1.
4. The introduction is also rewritten accordingly. In the revised version, we discuss the different situations of the axion insulator protected by inversion and time-reversal symmetries, describe the current unsolved issue in the community and our achievement on the prediction of spectral signatures of surface anomalous Hall effect.
5. We have improved the scientific writing throughout the whole paper to avoid the confusions raised by the referees. To sharpen our main contributions, we have slightly simplified the main text before Section “*Spectral signatures of the surface AHC – hinge states*”, and put the previous Fig. 4 to Supplementary Information.
6. Due to the contribution of the discussion on the signature of nonlocal transport measurements, we have added Prof. Haizhou Lu as a co-author.
7. Most of the referee comments/suggestions are implemented into the main text. Please see the itemized response below.

Comments of Referee #1 and authors’ reply:

1.1 Referee: *Mingqiang Gu et al. investigate the anomalous Hall effect in different magnetic topological phase such as axion, magnetic Weyl, and 3D Chern insulator, by using $k.p$ model. In addition, they take $\text{MnBi}_2\text{nTe}_{3n+1}$ and $\text{Mn}_2\text{Bi}_2\text{Te}_5$ as the examples to support the results of their $k.p$ model. Since 3D Chern insulator has not been discovered yet in real materials, authors try to realize this phase via tuning the hopping parameters in MnBi_2Te_4 . In the second part, authors report the real space distribution of anomalous Hall conductivity, especially focus on*

axion and Weyl phase. Finally, the authors show the real space surface and hinge states calculations of $MnBi_2Te_4$ to provide the possibility of experimental measurement of $0.5 e^2/h$.

(1) In this article, the author sometimes calls the FM $MnBi_{2n}Te_{3n+1}$ axion and sometimes the AFM $MnBi_{2n}Te_{3n+1}$ is axion, which makes the article difficult to read. It is recommended that the authors explain this part clearly. In addition, what is the difference between the axion state of $MnBi_{2n}Te_{3n+1}$ and the AFM sandwich structure FM(up)/TI/FM(down)?

Reply: If either time-reversal (T) or inversion symmetry (I) is present, the magnetoelectric coupling phase angle θ must be quantized as 0 or π (modulo 2π), the latter of which with an energy gap at the surface is also known as “axion insulators” defined in 3D systems. Since $MnBi_2Te_4/(Bi_2Te_3)_n$ in both FM and AFM phases have inversion symmetry, they are all I -preserved axion insulators.

Both $MnBi_2Te_4/(Bi_2Te_3)_n$ and FM(up)/TI/FM(down) are axion insulators with quantized bulk magnetoelectric coupling coefficient and half-quantized surface anomalous Hall conductivity $e^2/2h$. However, they differ from each other by the specific symmetry. $MnBi_2Te_4/(Bi_2Te_3)_n$ compounds are I -preserved axion insulators, while FM(up)/TI/FM(down) configurations are T -preserved axion insulators with magnetization-induced surface gaps because the bulk is nonmagnetic topological insulators.

We have added the following clarification on Page 2, Paragraph 2: “3D T -preserved topological insulators (TIs) possess the $\theta = \pi$ condition. However, the resulting half-quantized surface AHC is totally compensated by the gapless surface Dirac cones of TIs⁸. Therefore, the T -preserved axion insulator phase was typically realized by introducing extrinsic magnetic dopants to the top and bottom surfaces of a 3D TI to gap the surface states⁹⁻¹², while keeping the bulk still nonmagnetic. On the other hand, bulk magnetic TIs with inversion symmetry are categorized as I -preserved axion insulators¹³. Examples include the recent discovered superlattice-like stoichiometric compounds $MnBi_2Te_4/(Bi_2Te_3)_n$ in either ferromagnetic (FM) and antiferromagnetic (AFM) phases¹⁴⁻¹⁹.”

1.2 Referee: (2) In the $k.p$ model, authors assume the FM exchange field in all layers are the same, and adjust the parameter “ M ” to present different $MnBi_{2n}Te_{3n+1}$ ($n=1\sim 2$). For example, $MnBi_4Te_7$ ($M=1/2$) and $MnBi_2Te_4$ ($M= 1$). In real material of $MnBi_4Te_7$, there are two different species of unit blocks, $MnBi_2Te_4$ (septuple layer, SL) and Bi_2Te_3 (quintuple layer, QL). I am wondering why the Bi_2Te_3 cannot be regarded as a buffer layer in the $k.p$ model, but as having the same exchange field as $MnBi_2Te_4$.

Reply: We agree with that in real materials such as $MnBi_4Te_7$, the exchange fields in distinct van der Waals layers are indeed inhomogeneous. In the manuscript, our model Eq. (1) describes the phase diagram obtained by tuning the effective FM exchange field inside a 3D magnetic topological insulator, based on the assumption that the exchange field is homogeneous. We

believe that such assumption is valid to $\text{MnBi}_2\text{Te}_4/(\text{Bi}_2\text{Te}_3)_n$ because in such compounds the low-energy physics around the Fermi level is dominated by the p orbitals of Bi and Te, whereas the main role of Mn^{2+} (d^5) is to provide an exchange field. Such assumption works well for the mapping of $\text{MnBi}_2\text{Te}_4/(\text{Bi}_2\text{Te}_3)_n$ slabs to the 2D topological phase diagram [see PRL 123, 096401 (2019)]. With such assumption, here we are able to map 3D $\text{MnBi}_2\text{Te}_4/(\text{Bi}_2\text{Te}_3)_n$ materials with distinct integer n to one phase diagram (Fig. 1) by simply rescaling the density of Mn layers, instead of dealing with more complicated model Hamiltonians constructed by MnBi_2Te_4 and Bi_2Te_3 layers individually.

We have added the following clarification on Page 2, Paragraph 2: “*The mapping follows two assumptions: (i) the exchange field in different $\text{MnBi}_2\text{Te}_4/(\text{Bi}_2\text{Te}_3)_n$ materials is homogeneous; such assumption works well previously for the mapping of $\text{MnBi}_2\text{Te}_4/(\text{Bi}_2\text{Te}_3)_n$ slabs to the 2D topological phase diagram²⁵.*”

1.3 Referee: (3) *I am wondering why the homogeneous spin alignment do not contribute to the AHC, while inhomogeneous spin will provide the AHC. In addition, in Fig. 2(d) and (e), I am wondering why profile of MnBi_2Te_4 and MnBi_4Te_7 are quite different. For example, in FM phase, the AHC in MnBi_2Te_4 shows a 0.5 plateau and reach $C=1$ as $l=16$, while MnBi_4Te_7 shows an oscillation and does not increase significantly as $l=16$.*

Reply: In Fig. 2(d), the FM MnBi_2Te_4 shows a plateau at 0.5 while the AFM MnBi_2Te_4 is oscillating around 0.5. It is because we use every VdW layer for the partition of the local Chern marker in the real space. Recall that the in axion insulators the bulk contributes zero local Chern marker up to full unit cell, i.e., $\sum_{l \in \text{internal}}^{l+\text{u.c.}} C(l) = 0$. In MnBi_2Te_4 , Two VdW layers make an AFM unit cell while one VdW layer makes an FM unit cell. Therefore, the oscillation for the AFM phase can be regarded as incomplete integration within one unit cell. In fact, even for the case of FM, if one compute the integrated local Chern number with the contribution of a part of a unit cell, e.g., by separating one VdW layer into (Te-Bi-Te), Mn, and (Te-Bi-Te) sub-layers, oscillation behavior occurs as well. This argument also works for MnBi_4Te_7 , in which the oscillation period is two (four) VdW layers for the FM (AFM) configuration, as shown in Fig. 2(e) and Supplementary Fig. 2(d).

We have the relative discussion at Page 8, Paragraph 1: “*Overall, the locality of the surface AHC does not rely on the metallicity of the whole slab, but is due to the vanishing contribution of the local Chern marker from the bulk state, i.e., $\sum_{l \in \text{internal}}^{l+\text{u.c.}} C(l) = 0$.*”

The reason why MnBi_4Te_7 does not increase significantly to 1 as $l = 16$ is because the Fermi level cuts the conduction band of the Bi_2Te_3 termination (i.e., the 16th layer). Hence, the surface Chern marker is not well defined at this gapless termination. Details are shown in Supplementary Note 3 and Supplementary Fig. 2.

Supplementary Figure 2. (a) The electronic band structure for FM-MnBi₄Te₇. The horizontal lines denote different positions of the Fermi energy. (b) The integrated Chern number computed for FM-MnBi₄Te₇. Different curves correspond to different E_f denoted in (a). (c) The electronic band structure for a symmetric slab made of AFM-MnBi₄Te₇. Both surfaces are Bi₂Te₃-terminated, shown in the inset. (d) The integrated Chern number computed for the symmetric AFM-MnBi₄Te₇ slab.

1.4 Referee: (4) In the 2nd paragraph on p.6, authors mention the FM MnBi₂Te₄ turns into 3D Chern insulator as reducing the hopping parameter between the MnBi₂Te₄ bilayer. In this phase, the Chern number of a 2D slice within the k_x - k_y plane at an arbitrary k_z equals to 1. Authors also mention, in this phase, each bilayer contributes to an exact quantized Chern number 1. This result implies that 3D Chern insulator can be view as 2D Chern insulator stacking along z direction. However, monolayer MnBi₂Te₄ is trivial insulator, not 2D Chern insulator. I am wondering if author can explain more clearly in this part.vv

Reply: The referee is correct in that monolayer MnBi₂Te₄ is a trivial insulator, while FM bilayer MnBi₂Te₄ is a 2D Chern insulator. This is also confirmed by the previous DFT calculations [PRL 122, 107202 (2019); PRL 123, 096401 (2019)]. Similarly, monolayer Bi₂Te₃ is a large gap trivial insulator, while bilayer Bi₂Te₃ is a 2D topological insulator, much easier to be tuned to a Chern insulator. Therefore, the reason why we use bilayer, instead of monolayer, as the unit building block to build our phase diagram is to reach the 3D Chern insulator phase with a moderate FM exchange field, i.e., the effective FM exchange in MnBi₂Te₄.

In such context, the referee is also correct in that 3D Chern insulator can be viewed as 2D Chern insulator stacking along the z direction with weak enough coupling between 2D Chern insulators, as implied by Fig. 1. We added the following clarification to Page 4, Paragraph 3: “When the exchange field is strong enough, a 3D Chern insulator phase with chiral side surface states emerges, which is adiabatically connected to a vertical stacking of 2D Chern insulators³⁸⁻⁴⁰”

1.5 Referee: (5) In the p.7, line 198-200, “Firstly, while the axion insulator phase cannot be fully characterized by $\mathbb{Z}_4 = 2$, it manifests a surface AHC saturated at or oscillates around $e^2/2h$.” I am confused by this sentence, because in p.3, authors use \mathbb{Z}_4 to identify axion phase.

Reply: Since the symmetry indicator \mathbb{Z}_4 is not our main focus, we have deleted the confusing statement in the introduction.

$\mathbb{Z}_4 = 2$ is widely used to characterize an axion insulator phase in the literature, e.g., *Sci Adv* **6**, eaba4275 (2020), *PRL* **124**, 136407 (2020), *PRB* **102**, 035144 (2020). However, we find that a distinct topological phase in our phase diagram, i.e., 3D Chern insulator phase, also manifests $\mathbb{Z}_4 = 2$. Therefore, we clarify this point at Page 5, Paragraph 1: “The latter corresponds to an axion insulator phase, as predicted by previous studies using a single \mathbb{Z}_4 invariant^{19, 26, 47}. Nevertheless, we find that the 3D Chern insulator phase also yields $\mathbb{Z}_4 = 2$ but a different parity distribution compared with the axion insulator.”

1.6 Referee: (6) In the 2nd paragraph on p.9, authors propose that there is a hinge state on the top surface but there is no hinge state on the bottom surface. They mention the domain wall of this system as a higher-order TI. I am confused by this paragraph. In HOTI, the hinge state will go around the edge of 3D system obey the crystal symmetry. It means, if three directions of system are finite, we start from the top surface and moving along the hinge state, we will reach the bottom surface via the side edge, and finally return to the upper surface, complete a 1D closed loop. I am wondering if author can explain more clearly in this part.

Reply: We apologize for the ambiguous description on the HOTI that confused the referees (Referee #2 also pointed out this). For FM MnBi_2Te_4 axion insulator phase, we want to compare the features of the in-band hinge state (hall mark of surface anomalous Hall and axion phase, see Page 8, Paragraph 3) with the in-gap hinge state, which is the hallmark of HOTI phase. We are focusing on a specific side surface, resulting in two hinge states with the top and bottom surfaces. On the other hand, the specific circulation of the HOTI in-gap hinge mode depends on the geometry of the sample. For example, if the geometry is a hexagonal prism, the hinge mode cycles around the whole sample; if the geometry is a trigonal prism, the hinge mode cycles at the top or the bottom surface, as shown in the following sketch:

Hexagonal prism

Trigonal prism

We have rewritten the whole section “Spectral signatures of the surface AHC – hinge states” from Page 8 and replotted the schematic device plot in Fig. 3a. Especially, since we are focusing on the in-band hinge states rather than the HOTI in-gap hinge state, we added the following clarification on Page 9 Paragraph 2: “In such a situation, the in-band hinge states still exist, with the top and bottom hinges having the same chirality, as shown in Fig. 3e and 3f. In addition, such a side surface exhibits a half-quantized local Chern marker, leading to a single chiral mode ($\frac{1}{2} + \frac{1}{2}$) at the top hinge and no chiral modes ($\frac{1}{2} - \frac{1}{2}$) at the bottom hinge, inside the side surface gap (position ② and ④, see Fig. 3f). Such a chiral hinge mode with integer AHC, denoted as in-gap hinge mode, is equivalent to the chiral domain wall state (also see Supplementary Note 5) in high-order topological insulators^{57, 58}, which is also predicted in FM $\text{MnBi}_2\text{Te}_4/(\text{Bi}_2\text{Te}_3)_n$ by model calculations²⁶. However, we note that such in-gap hinge mode only exists within a small energy range, i.e., 6 meV for FM MnBi_2Te_4 because its side surface gap is a high-order magnetization gap. On the other hand, the in-band hinge modes for both of the top and bottom hinge, originated from the half-quantized surface AHC, remain robust within a much larger energy range, i.e., the top/bottom surface gap (52 meV), which is favorable for experimental detection.”

1.7 Referee: (7) Fig.2, the caption does not specify the number of layers represented by different color lines. (b) and (c) are incorrectly marked.

Reply: We have fixed the typo in the figure caption, and added the following clarification to the caption of Fig. 2: “In panel (a-c), the color lines terminate at different positions, denoting the total thickness of the calculated slabs up to 16 VdW layers.”

1.8 Referee: (8) Fig. 3(a) is the side surface state of AFM MnBi_2Te_4 . Previous work reports the ground state of AFM MnBi_2Te_4 is an AFM-TI. I am wondering why the Dirac point in Fig.3(a) is not on the Gamma point. The result of Fig.3(a) seems like a topological crystalline insulator.

Reply: We thank the referee for pointing out our overlook in the previous version by putting the calculation of the FM phase as Fig. 3a (the main conclusion summarized in schematic Fig. 3f, now Fig. 3b, is still correct). Now we have replaced it with the correct plot for the AFM phase [new Fig. 3c spot ③]. It is shown that the Dirac point of the side surface state of AFM MnBi_2Te_4 is right at the Gamma point, as expected.

Comments of Referee #2 and authors' reply:

2.1 Referee: In this manuscript, the authors provided a systematic study on the surface anomalous Hall conductance (AHC) for the recently discovered topological magnetic materials $\text{Mn}(\text{Bi},\text{Te})$. Based on the first-principles-based calculations of local Chern marker, they managed to extract the AHC for each layer in a finite-size slab of various magnetic topological phases, and summarized several interesting features for the AHC. I think the current paper contains some interesting results that might be publishable. However, some results and statements seem quite confusing and likely NOT valid. Meanwhile, I find the current draft hard to read and the authors should spend lots of efforts to improve the readability, including fixing the typos and refining logical flow in writing. In addition, I am not convinced that the novelty level of this work satisfies the high standards of Nature Comm., given that some results might have already existed in the literature. Therefore, I cannot recommend this work for publication in Nature Comm.

Reply: We thank the referee for reading our manuscript, and understand in some extent the referee's concern about the novelty. In the revised version, besides the multiple topological phases and the surface AHC calculated in the atomistic DFT level, we also proposed a key contribution to the community, i.e., a clear prediction of the unique transport properties in the experiment. Please also see the summary of the main changes and the itemized reply in the following.

Therefore, we believe that the current version contains enough novelty that distinguishes all the existing literature and meets the stringent standards of high technical quality and scientific rigor of *Nature Communications*.

2.2 Referee: Here're some comments to support my judgement:

1. While the authors provide a detailed ab-initio-based studies on the surface AHC of a specific family of materials, the main conclusion they have reached, which I believe is the surface localization of AHC, do not surprise me at all. For example, it has been a common sense in the community that axion insulators generally host half-quantized surface AHC, which, by definition,

should be localized around the surface. This surface localization of AHC can be intuitively understood by considering a magnetic domain wall on the surface of a 3D TI, which is known to host a chiral domain wall mode. Since such chiral mode signals the sign change of half-quantized surface AHC, its localization around the surface should also implies the localization of surface AHC.

Reply: We agree with the referee that the half-quantized AHC hosted by an axion insulator is a common sense in the community, which we have pointed out in the first paragraph in the introduction part. However, the features of surface AHC in a realistic material, e.g., the localization length, the profile of the surface AHC as a function of the penetration depth, and the effect of metallic bands, are much more complicated. For example, the in-gap hinge state is also expected to be localized at the hinge. However, our calculation shows that the in-gap hinge state decays rapidly along the horizontal (x) direction at the top surface, while it decays much more slowly along the vertical (z) direction at the side surface. This seems counterintuitive due to the weak VdW interaction between layers along the z direction (see Page 9, Paragraph 3 for the relevant analysis). Therefore, it is important to construct atomistic Hamiltonians from first-principles calculations with close reliance on realistic materials, instead of using over-simplified models.

More importantly, how does a half-quantized surface AHC look like in the axion insulator phase, and what is the experimental signatures of such surface AHC are also unsolved problems for a long time. In the revised manuscript, we have elaborated a lot more to unveil these puzzles, including the new Fig. 3, Fig. 4 and the rewritten section of “*Spectral signatures of the surface AHC – hinge states*” from Page 8 to Page 10. We find that the surface AHC is represented by the chiral hinge mode embedded inside the side surface bands, providing guidelines for nonlocal transport measurements. Given that the previous signature of axion insulators focuses on zero Hall plateau that cannot distinguish with that of a trivial insulator, our study for the first time provides direct evidence to detect the axion insulator phases. Please see the main text as well as the following itemized response for the details.

2.3 Referee: 2. It seems to me that authors failed to acknowledge some contributions in the previous literature, which is somewhat disappointing. In particular, as far as I understand, part of the results in this work seems to have some overlaps with the model study in Ref. 27, which, however, is only mentioned very briefly in the introduction. Below, I list a few examples for the authors to think about:

(i) The Z4 indicator for ferromagnetic axion insulating phase in Mn(Bi,Te) was simultaneously and independently calculated in the following references A & B (see below) as well as in Ref. 27.

[A] Hu, Chaowei, et al. "Realization of an intrinsic ferromagnetic topological state in MnBi₈Te₁₃." Science advances 6.30 (2020): eaba4275.

[B] Tian, S., Gao, S., Nie, S., Qian, Y., Gong, C., Fu, Y. & Shin, S. (2020). *Magnetic topological insulator MnBi₆Te₁₀ with a zero-field ferromagnetic state and gapped Dirac surface states*. *Physical Review B*, 102(3), 035144.

Reply: Our statement is that $\mathbb{Z}_4 = 2$ is not a sufficient condition to identify an axion insulator, because 3D Chern insulator could also manifest $\mathbb{Z}_4 = 2$. That is the reason why we did not directly compare with the previous calculations claiming axion insulator phases just because $\mathbb{Z}_4 = 2$. By the way, we declare that Reference A pointed by the referee was already cited twice in the previous manuscript (old Ref. 20).

Per referee's request, we have added the following clarification to Page 5, Paragraph 2: "*The latter corresponds to an axion insulator phase, as predicted by previous studies using a single \mathbb{Z}_4 invariant^{19, 26, 47}. Nevertheless, we find that the 3D Chern insulator phase also yields $\mathbb{Z}_4 = 2$ but a different parity distribution compared with the axion insulator.*"

2.4 Referee: (ii) *Can the authors explain the difference between the effective Hamiltonian in Ref. 27 and the one in Eq. 1 for this work? I find these two Hamiltonians quite similar to each other, and I think the authors should address the similarities and/or discrepancies in the main text, when Eq. 1 first shows up.*

Reply: We disagree with the argument that the two Hamiltonians are quite similar to each other. The differences between them are very obvious, and can be mainly summarized by two aspects: 1) starting point; 2) functionality. Let me explain in the following:

- 1) The effective Hamiltonian in Ref. 27 starts from a **3D bulk TI**, which is obtained by [Nat. Phys. 5, 438 (2009); PRB 82, 045122 (2010)]. On the other hand, our Hamiltonian Eq. (1) starts from the **2D monolayer** Bi₂Te₃ and other buffer layers to build a 3D FM topological system via layer construction, which is mainly inspired by Ref. 35 (Ref. 41 in the new version). The main difference between our model and Ref. 35 is that we use the phase diagram to identify the axion insulator phase (as well as the fragile QSH insulator phase, both of which are not mentioned in Ref. 35), which is the main focus of our paper.
- 2) The purpose of the Hamiltonian in Ref. 27 is to investigate the high-order TI phase in MnBi₂Te₄/(Bi₂Te₃)_n with various magnetic configurations, while ours is to study more 3D topological phases. Therefore, the treatments of the two models are also different. Ref. 27 considers different magnetic moments for different sublayers, while ours assumes a homogeneous FM exchange but considers buffer layers to tune the inter-bilayer coupling. From our layer-construction perspective, topological phase transitions are succinctly realized by changing the inter-bilayer hopping.

By the way, we want to comment that the similarity between the model from Ref. 27 and our previous work [PRL 123, 096401 (2019)] is much more significant. The starting points of those two works (i.e., a 3D bulk TI) are the same. The only difference is that Ref. 27 considers more

magnetic configurations, while [PRL 123, 096401 (2019)] assumes a 2D FM phase to study the potential QAH effect. Given that [PRL 123, 096401 (2019)] is not even cited by Ref. 27, we do not think this comment is fair enough and appropriate.

2.5 Referee: (iii) *As far as I am aware of, Ref. 27 appears to be the first prediction of chiral hinge modes in FM Mn(Bi, Te). Even though the hinge mode calculation is regarded as an important result of the current paper, the contribution of Ref. 27 is not mentioned at all. Can the authors explain the similarities and differences between their hinge mode calculations with those in Ref. 27?*

Reply: We agree with the referee that Ref. 27 firstly presented the higher-order topology in FM $\text{MnBi}_2\text{Te}_{3n+1}$. In fact, the hinge mode pattern of the FM topological insulator had been pointed out earlier in certain references [PRB 97, 155305(2018); Nat. Phys. 15, 577(2019)], especially in magnetic axion insulator EuIn_2As_2 [PRL 122, 256402(2019)].

About the similarities and differences, we have rewritten the whole section “*Spectral signatures of the surface AHC – hinge states*” from Page 8 to distinguish two different kinds of hinge modes, i.e., in-band hinge and in-gap hinge modes, the latter of which corresponds to the one predicted in Ref. 27 in the FM case. The most important results of our work is the unique in-band hinge states as the spectral signature of the magnetic axion insulators (no matter FM and AFM), while in-gap hinge states are discussed in comparison.

We have added the clarification and the citation of Ref. 27 (now Ref. 26) on Page 9 Paragraph 2: “*In such a situation, the in-band hinge states still exist, with the top and bottom hinges having the same chirality, as shown in Fig. 3e and 3f. In addition, such a side surface exhibits a half-quantized local Chern marker, leading to a single chiral mode ($1/2 + 1/2$) at the top hinge and no chiral modes ($1/2 - 1/2$) at the bottom hinge, inside the side surface gap (position ② and ④, see Fig. 3f). Such a chiral hinge mode with integer AHC, denoted as in-gap hinge mode, is equivalent to the chiral domain wall state (also see Supplementary Note 5) in high-order topological insulators^{57, 58}, which is also predicted in FM $\text{MnBi}_2\text{Te}_4/(\text{Bi}_2\text{Te}_3)_n$ by model calculations²⁶. However, we note that such in-gap hinge mode only exists within a small energy range, i.e., 6 meV for FM MnBi_2Te_4 because its side surface gap is a high-order magnetization gap. On the other hand, the in-band hinge modes for both of the top and bottom hinge, originated from the half-quantized surface AHC, remain robust within a much larger energy range, i.e., the top/bottom surface gap (52 meV), which is favorable for experimental detection.”*

2.6 Referee: *In section “Surface AHC of magnetic topological phases”, the authors mentioned that the FM axion insulating phase is found to have a saturation of Chern number quantized to one, as the sample thickness increases. This is exactly the higher-order TI phase proposed in Ref. 27, am I correct?*

Reply: We disagree with that. In the context of our manuscript, mentioning slabs of a FM axion insulator as a high-order TI is neither necessary nor accurate. The explanation is in the following:

1) For an I -preserved axion insulating phase, if we consider a slab where I is still preserved, e.g., FM or odd-layer AFM slabs, the total Chern number reaches to 1 because both of the top and bottom surfaces contribute half quantization with the same sign. Such $(\frac{1}{2} + \frac{1}{2})$ scenario, as mentioned in the introduction part, is well known for a 3D TI with the same magnetization at both top and bottom surfaces [e.g., see the review paper SciPost Physics 6, 046 (2019)]. In contrast, high-order TI manifest a $C = 1$ chiral mode on one surface and no chiral modes on the other side (i.e., $1 + 0$), which is different from that of an axion insulator phase.

2) We also note that the Chern number is discussed in the context of a 2D topological system, such as a slab of a 3D axion insulator. In contrast, high-order TI is 3D TI phase with 1D hinge states, whose circulation depends on the signs of magnetization of the side surfaces. They are conceptually different.

We speculate that the referee may want to imply that the FM slab in our calculation has the same edge state as the hinge mode predicted in Ref. 27, but they are still fundamentally different. The hinge modes appearing in Ref. 27, e.g., Fig. 4c, requires that the side-surface magnetization gap (~ 6 meV) originated from warping effect overwhelms the finite size effect (the x and y direction should be finite), while a Chern insulator only requires a larger magnetization gap at the top surface (~ 50 meV) than the finite size gap (the z direction should be finite).

2.7 Referee: 3. The authors defined a 3D fragile QSH insulator and claimed it to have fragile topology without providing any supporting evidence. To me, the concept of fragile topology has a rigorous definition. Namely, a fragile topological insulator by itself cannot be described by symmetric Wannier functions, while such a Wannier obstruction is “fragile” and can thus be removed upon coupling to a set of trivial atomic bands. Therefore, the authors will need to prove the fragile topological nature based on the above definition before making such claims.

Reply: We did not define the fragile phase, but attribute the 3D QSH insulator to the fragile topological phase. As the referee said, the rigorous definition of the fragile topology indicates that the phase by itself cannot be described by symmetric Wannier functions, while such a Wannier obstruction is “fragile” and can thus be removed upon coupling to a set of trivial atomic bands. To verify the fragile topology one can analyze the symmetry eigenvalues of inversion and the minimal atomic limits, i.e., elementary band representation. Recall the assumptions of our model Hamiltonian that the band inversion only occurs at $\Gamma = (0,0,0)$ and $Z = (0,0,\pi/D)$, which means that the band orders at the other inversion-invariant momenta remain the same as that of the nonmagnetic 2D limit, i.e., bilayer Bi_2Te_3 . We note that the parities shown in Fig. 1 for a 2D T-broken QSH insulator (two “-”s at Gamma and two “+”s at the other points) is already sufficient to ensure a nontrivial topological phase (including the fragile topology). On the

other hand, the phase diagram clearly distinguishes such a phase from the axion insulator phase and 3D Chern insulator phase. This means that this phase corresponds to a fragile topological phase.

Since fragile topology is not our focus here, we added the following simplified clarification to Page 5, Paragraph 2: “*The former, i.e., 3D QSH insulator, is equivalent to the vertical stacking of 2D T-broken QSH insulators with the parities shown in Fig. 1. Such a phase cannot be described by symmetric Wannier functions, while the Wannier obstruction can be removed upon adding a set of trivial elementary band representations. Therefore, it corresponds to a distinct type of “fragile topology”, manifesting a novel twisted bulk-boundary correspondence^{45, 46}.*”

2.8 Referee: 4. I find the last paragraph of the section “Experimental realization of surface AHC” rather confusing. The authors conclude that for the FM axion insulating state, the chiral hinge mode only circulates around the top surface, but not the bottom surface. This does not make sense to me, since the FM state does have a spatial inversion symmetry (based on which the Z4 indicator is calculated).

In fact, the absence of hinge mode for the bottom hinge can be naturally explained by Fig. 4c of Ref. 27, even considering inversion symmetry. As shown in Fig. 4c of Ref. 27, both C3 and inversion symmetries enforce a special pattern of the chiral hinge mode around the sample, such that for a given side surface, the hinge mode is always absent on either the top or the bottom hinge. In this regard, I think the authors have falsely assumed that a uniform magnetic domain wall is formed between the top surface and every side surface, as they discussed in Supplementary Note 5. In fact, as pointed by Ref. 27 in the appendix, it is the hexagonal warping term in Mn(Bi, Te) that makes the domain wall to appear only in alternating side surfaces, which causes the hinge mode pattern in Fig. 4c.

To clarify the hinge mode pattern, I would suggest the authors to calculate not only the top & bottom hinge modes on the same side surface, but also those on a parallel side surface on the back of the sample. I believe for the other side surface, the hinge mode can only appear on the bottom hinge, according to the Fig. 4c of Ref. 27.

If the hinge mode pattern is indeed the one in Fig. 4c of Ref. 27, does the proposed non-local transport setup still work?

Reply: We apologize for the ambiguous description on the HOTI that confused the referees (Referee #1 also pointed out this). For FM MnBi₂Te₄ axion insulator phase, our purpose is to compare the features of the in-band hinge state (hall mark of surface anomalous Hall and axion phase, see Page 8, Paragraph 3) with the in-gap hinge state, which is the hallmark of HOTI phase predicted by Ref. 27. Therefore, we are not considering the relationship of multiple side surfaces, but focusing on a specific side surface, resulting in two hinge states with the top and bottom surfaces. Also, we do not find from our Supplementary Note 5 the assumption of a uniform

domain wall between the top surface and every side surface. The purpose of Supplementary Note 5 is to elucidate that a $C = 1$ in-gap hinge mode can be considered as the domain wall of two surface anomalous Hall effect with Chern numbers $\frac{1}{2}$ and $-\frac{1}{2}$.

The specific circulation of the HOTI in-gap hinge mode depends on the geometry of the sample. For example, if the geometry is a hexagonal prism, with inversion symmetry, the hinge mode cycles around the whole sample as predicted in Ref. 27; if the geometry is a trigonal prism, the hinge mode cycles at the top or the bottom surface, as shown in the sketch:

About the non-local transport setup for FM MnBi_2Te_4 , distinct from the in-gap hinge states, the in-band states are buried in the surface side surface bands, thus requiring nonlocal detection that distinguishes the left-moving and right-moving modes, as well as the Fermi level cutting the surface bands. In comparison, the in-gap hinge modes which is the hallmark of the high-order TI phase (e.g., the one predicted in Ref. 27), could be detected by the typical two-terminal transport measurements when the Fermi level is inside the surface band gap, because there is a single mode carrying a quantized Chern number (see Fig. 3e). We have added such conclusion on Page 10, Paragraph 2: “*We thus propose a device setup with a thick-enough sample and multi-terminal leads attached to one surface covering only a few VdW layer. While the signal of in-band hinge modes is buried by the metallic side surface states for typical two-terminal transport measurements, one can expect nonzero signal through nonlocal surface transport measurements*⁶⁰.”

2.9 Referee: 5. *While reading the paper, I am often confused about which of the results are obtained via first-principles calculations and which are from the effective Hamiltonian. In addition, most discussions have involved both AFM phase and FM phase. Sometimes, I could easily get lost while reading a specific statement, being not clear about which phase the statement is referred to. I would suggest the authors to improve the writing of the draft and makes it easier for readers to follow.*

Reply: We apologize for the confusion. In the revised version, only the phase diagram shown in Fig. 1 is obtained by the effective Hamiltonian Eq. (1), while all the other calculations, including

the electronic structure, the \mathbb{Z}_4 symmetry indicator, local Chern marker, surface anomalous Hall conductivity and the local density of states of the surface and hinge states of $\text{MnBi}_2\text{Te}_4/(\text{Bi}_2\text{Te}_3)_n$, are obtained by DFT calculations. Some properties among them are calculated by tight-binding models with Wannier basis transformed from DFT-calculated Bloch wavefunctions. We clarified this in the Methods section. We have also changed the title of the second section to “*Multiple topological phases from model Hamiltonian calculations*”, the third section to “*DFT-calculated Surface AHC of magnetic topological phases in $\text{MnBi}_2\text{Te}_4/(\text{Bi}_2\text{Te}_3)_n$* ”.

We have also improved the writing of the whole draft, including clear statements when involving both FM and AFM phases.

2.10 Referee: 6. I don't quite understand the physical origin of the chiral hinge mode that coexists with the gapless surface state in the AFM phase. Can the authors elaborate on this?

Reply: We have rewritten the whole section “*Spectral signatures of the surface AHC – hinge states*” and proposed that the in-band hinge states are the central spectral signature of magnetic axion insulator phases. Please see the revised Fig. 3 and Page 8, Paragraph 3 for the AFM case: “*The calculated LDOS of the top hinge and side surface states of AFM MnBi_2Te_4 are shown in Fig. 3c (position ② and ③). By comparison, we find that a remarkable feature of the hinge state is the asymmetric spectral weight between the left and right-moving modes, indicating its chiral nature. This can be understood by the chiral top surface AHC embedded into the helical gapless side surface states. The top, bottom and side surface states, as well as the top and bottom hinge states are schematically shown in Fig. 3b. We denote such chiral hinge modes embedded inside the side surface bands as “in-band hinge” states. For even-layer AFM slabs where the top and bottom surfaces have opposite magnetizations, the top and bottom in-band hinge states manifest opposite chiralities accordingly.*”

We also compare the in-band hinge states and in-gap hinge states (the hallmark of HOTI phase) in FM MnBi_2Te_4 , please see the new Fig. 4 and the relevant discussion in Page 9 Paragraph 2 and 3.

2.11 Referee: 7. In the last paragraph of section “Multiple topological phases...”, the authors mentioned “any transition between two of the three insulating phases must be connected by an intermediate Weyl-semimetal phase”. This is a very strong statement. I think it is generally possible (maybe in a different model) that the phases are separated by a line of topological phase transition, but not a finite region.

Reply: The conclusion is indeed based on our model as well as the parity analysis discussed in this section. We have now rephrased our observation in Page 5, Paragraph 3: “*In the phase diagram derived from Eq. (1), T symmetry is always broken except for the horizontal line with M*

= 0. We find that that the three magnetic insulating phases are isolated from each other by an intermediate Weyl-semimetal phase, which is also consistent with the parity analysis.”

2.12 Referee: 8. Can the authors provide the model parameters for Eq. 1?

Reply: Done. The model parameters and the analytical solutions of Eq. (1) are provided in Supplementary Note 1: “ $\hbar v_f = 2.6 \text{ eV} \cdot \text{\AA}$, $\Delta = 0.019 \text{ eV}$, $B = -40 \text{ eV} \cdot \text{\AA}^2$, $d_0 = 2.7 \text{ \AA}$, $t_{IB}^0 = 0.033 \text{ eV}$, and $\alpha = 4.6$ ”.

2.13 Referee: 9. I think there is a typo in the last two paragraphs of the section “Surface AHC of magnetic topological phases”, where Fig. 2b and Fig. 2c are messed up with each other.

Reply: Done. We have fixed the typo in the figure caption.

Comments of Referee #3 and authors’ reply:

3.1 Referee: In this manuscript, the authors studied the Hall effect and the quantization of the Hall conductance on the axion insulator's surface. A phenomenological model was constructed as stacked 2D TI layers with interlayer couplings and a ferromagnetic Zeeman field. This model has a rich phase diagram, including the axion insulator, the Weyl semimetal, the 3D Chern insulator, and a magnetic fragile topological insulator. This model well explains the various topological phases in the popular material family $\text{MnBi}_2\text{Te}_4/(\text{Bi}_2\text{Te}_3)_n$. The authors also applied DFT calculation to the materials $\text{MnBi}_2\text{Te}_4/(\text{Bi}_2\text{Te}_3)_n$ and computed the local Chern markers, from which they found that the top or bottom surface of an axion insulator is either 1/2 or -1/2. The +1/2 Chern markers are consistent with the theoretical prediction of half-quantized Hall conductance of a single surface of the axion insulator.

The manuscript is well written, and the calculations are solid. In light of the high standards of Nature Communication, I think a key aspect in the assessment of the present manuscript should be a clear prediction of the unique transport properties in the experiment. In my opinion, however, the proposed experiments are not well explained. Therefore, I can recommend publishing in Nature Communication if the authors could address the following comments satisfactorily in a revised version.

Reply: We thank the referee for appreciating the quality of our work and pointing out the key contribution that was somehow buried in the previous version, i.e., a clear prediction of the unique transport properties in the experiment. In the revised manuscript, we have elaborated a lot

more on this point, with the new Fig. 3, Fig. 4 and the rewritten section of “Spectral signatures of the surface AHC – hinge states” from Page 8 to Page 10. We also changed the title to “Spectral signatures of the surface anomalous Hall effect in magnetic axion insulators”. Please see the summary of the main changes and the itemized reply in the following.

3.2 Referee: (1) In the section “Half-quantized surface AHC in axion insulator phases” and in Figure 2, it is shown that the FM axion insulator has a total Chern number 1 whereas the AFM axion insulator has a total Chern number 0. My question is whether this statement depends on the slab's configuration, e.g., the even/odd layers' effect. From the aspect of inversion symmetry, which stabilizes the axion phase, there is no crucial difference between the FM and AFM axion insulators. In general, I believe that the total Chern number of an axion insulator must be ± 1 if the sample respects the inversion symmetry. From Supplementary Figure 2, the AFM slab does break the inversion symmetry. I wonder whether an additional bottom layer with spin pointing up will recover the inversion symmetry and make the total Chern number odd.

Reply: The referee is correct. For the odd number AFM phase with nonzero net magnetic moment, the total Chern number could reach 1 if the net magnetization is strong enough, which is the case of $\text{MnBi}_2\text{Te}_4/(\text{Bi}_2\text{Te}_3)_n$ thick slabs. To confirm this, we have computed the AFM slab with odd number of magnetic layers, so that the top and bottom layer has the same spin direction. The following figure shows the Integrated local Chern marker $C(l)$ for a 17-layer MnBi_4Te_7 slab with inversion symmetry, which clearly shows $C = 1$ for the whole slab.

We have added the following clarification to Page 7, Paragraph 2: “For slab calculations, the total Chern number depends on whether I is preserved in the slab geometry. Therefore, for FM and odd-layer AFM MnBi_2Te_4 slabs with I , the total Chern number reaches to 1 because both top and bottom surfaces contribute the same AHC $e^2/2h$. On the other hand, as shown in Fig. 2(d), the topmost layer of the even-layer AFM phase (with broken I) contributes almost half-quantized

AHC, i.e., $C_z(1) = 0.49$, while the bottom layer contributes an opposite AHC $C_z(16) = -0.49$, achieving a zero Hall plateau state and a zero total Chern number.”

3.3 Referee: (2) *The local Chern markers do show half-quantization around the surfaces. However, it is known that it is not clear how this half-quantization is reflected in the experiment. In the section "Experimental realization of surface AHC", the authors discussed two different cases: (a) the side surface is almost gapless (b) the side surface is gapped. Case (a) is not well explained. I think the sentence "One is the finite-size gap originated from the coupling between the top and bottom surfaces" should be "One is the finite-size gap originated from the coupling between the left (front) and right (back) surfaces" because the top and bottom surfaces are assumed as gapped in the first sentence in this paragraph. The finite size gap is exponentially small in a macroscopic or mesoscopic sample. So I am not sure whether the discussion based on the finite-size gap is relevant. Also, the statement "nonlocal transport signature of the chiral hinge state appears when the Fermi level crosses the side surface bands ..." is also confusing: what's the configuration of the chiral hinge states in real space? What kind of nonlocal transport signature? Why does the Fermi level need to cross the surface bands?*

Reply: Indeed, the discussion of the finite-size effect is not necessary here. In the revised version, we employed a new DFT-based method to compute the hinge states shown in Fig. 3 (see Methods). To eliminate the finite size effect, the hinge states are calculated using a bi-semi-infinite open boundary geometry condition. The structure is semi-infinite along the x - and z -directions, while the periodic boundary condition is maintained along the y -direction so that k_y remains a good quantum number.

We have now rewritten case (a) in Page 8 Paragraph 3, and concluded that the main feature of the hinge state is the asymmetric spectral weight between the left and right-moving modes, indicating its chiral nature. This can be understood by the chiral top surface AHC embedded into the helical gapless side surface states. Such chiral hinge modes embedded inside the side surface bands are denoted as “in-band hinge” states and discussed in both case (a) and (b) throughout this section.

The configuration of the in-band hinge in the real space is shown in Fig. 3b (top and bottom hinge). Distinct from the in-gap hinge states in case (b), such in-band states are buried in the surface side surface bands, thus requiring nonlocal detection that distinguishes the left-moving and right-moving modes, as well as the Fermi level cutting the surface bands. In comparison, the in-gap hinge modes, which is the hallmark of the high-order TI phase, could be detected by the typical two-terminal transport measurements when the Fermi level is inside the surface band gap, because there is a single mode carrying a quantized Chern number (see Fig. 3e). We have added such conclusion on Page 10, Paragraph 2: “*We thus propose a device setup with a thick-enough sample and multi-terminal leads attached to one surface covering only a few VdW layer. While the signal of in-band hinge modes is buried by the metallic side surface states for typical two-*

terminal transport measurements, one can expect nonzero signal through nonlocal surface transport measurements⁶⁰.”

3.4 Referee: (3) *In case (b), the side surface is gapped and contributes to an additional 1/2 or -1/2 AHC's to the top and bottom surfaces, leading to integer AHC on the top (1) and bottom (0) surfaces. My question is how to distinguish this top surface with AHC 1 from a 2D QAH on top of a trivial insulator? The two situations have the same surface AHC but different bulk topology.*

Reply: That is an interesting question. In the revised version, we calculated the hinge states projected onto different positions of the side surface (along the z direction, Figs. 4f-4j). As shown in Fig. 4j, although the in-gap hinge has $C = 1$ hinge mode at the top surface and no hinge at the bottom surface, the chiral in-band hinge state decays rapidly through both of the top and the side surfaces into the bulk. Such a unique profile is for sure distinct from that of a 2D Chern insulator on top of a trivial insulator.

We have added the following comparison in Page 10, Paragraph 1: *“On the other hand, the chiral in-band hinge state decays rapidly through both of the top and the side surfaces (green curves in Fig. 4e and 4j), which agrees well with the locality of the surface AHC shown in Fig. 2d. Such consistency again demonstrates that the in-band hinge state is an ideal physical quantity to verify the existence of the surface AHC of the axion insulator phase. Note that the in-band hinge profile in Fig. 4j also distinguishes that of a FM trivial insulator and a Chern insulator on top of a trivial insulator.”*

On the other hand, the top (1) and bottom (0) in-gap hinge modes have a unique response to the magnetic field. If the magnetic field reverses the spin orientation, the in-gap hinge mode moved from the top surface to the bottom surface. On the other hand, a 2D Chern insulator on top of a trivial insulator does not manifest such response under magnetic field.

3.5 Referee: (4) *The authors also mentioned that "Such measurements serve as the direct evidence of axion insulators, which is in sharp contrast to the case of a trivial insulator". For the paper to be more self-contained, it is better to explain such measurements and explicitly point out the sharp difference.*

Reply: We proposed that the sharp difference between the axion insulator and trivial insulator phase is the in-band hinge states that manifest the surface anomalous Hall effect. We have explained the relevant measurements and the sharp difference in the rewritten Section *“Spectral signatures of the surface AHC – hinge states”*. Please see the summary paragraph in Page 10: *“The in-band hinge states will contribute to unique transport signatures when E_f crosses the side surface bands. As shown in Fig. 3d, compared with non-chiral top and side surface states, these localized chiral in-band hinge states exhibit imbalanced spectral weight for the left-moving and*

right-moving modes. We note that even for FM axion insulators, one can probe a specific hinge with only in-band hinge contribution (position ④ in Fig. 3f). We thus propose a device setup with a thick-enough sample and multi-terminal leads attached to one surface covering only a few VdW layer. While the signal of in-band hinge modes is buried by the metallic side surface states for typical two-terminal transport measurements, one can expect nonzero signal through nonlocal surface transport measurements⁶⁰. Although the exact number of $e^2/2h$ conductance is not topologically protected and not immune from subtle device structure and disorder effects, the chiral in-band hinge states still give rise to unambiguous transport signature as the direct evidence of axion insulators, which is in sharp contrast to the case of a trivial insulator⁶⁰.”

3.6 Referee: *The followings are some minor comments.*

(5) *It would be good if the authors can clarify the symmetries that equation 1 has. For example, the inversion operator and the combined operation of time-reversal and translation in the AFM phase.*

Reply: Eq. (1) and the resulting phase diagram (Fig. 1) only describes FM configurations. We have added the following sentence to Page 4, Paragraph 2: “*The inversion symmetry $I = \tau_x \nu_x$ is preserved in this Hamiltonian.*”

3.7 Referee: (6) *Equation 1 does not have a Brillouin zone; hence equation 2 can not be directly applied. The authors must assume the parities at $(k_x, k_y) = (\pi, 0), (0, \pi), (\pi, \pi)$ for $k_z = 0$ and π . It is helpful to explain how this equation is evaluated.*

Reply: The referee is correct. To achieve the mapping between the symmetry indicators of different $\text{MnBi}_2\text{Te}_4/(\text{Bi}_2\text{Te}_3)_n$ phases and the phase diagram derived from Eq. (1), we have added the following assumptions to Page 5, Paragraph 1: “*The mapping follows two assumptions: ... (ii) the band inversion only occurs at $\Gamma = (0, 0, 0)$ and $Z = (0, 0, \pi/D)$, which means that the band orders at the other inversion-invariant momenta remain the same as that of the nonmagnetic 2D limit ($M = 0, D \rightarrow \infty$), i.e., bilayer Bi_2Te_3* ”

3.8 Referee: (7) *The 3D Chern insulator indeed has the same z_4 index as the axion insulator. However, this does not mean that the 3D Chern insulator cannot be distinguished from the axion insulator through the symmetry-based indicators. Considering the inversion symmetry, the indicator group is $Z_4 * Z_2 * Z_2 * Z_2$, where the Z_2 indicators can be chosen as the parity of the Chern number in the $k_i = \pi$ ($i = x, y, z$). It is known that when $z_2 = 1$, which is true in the 3D Chern insulator, $z_4 = 2$ no longer corresponds to the axion insulator.*

Reply: We thank the referee for the valuable comment. We have added the following clarification to Page 5, Paragraph 2: “*The latter corresponds to an axion insulator phase, as predicted by previous studies using a single \mathbb{Z}_4 invariant^{19, 26, 47}. Nevertheless, we find that the 3D Chern insulator phase also yields $\mathbb{Z}_4 = 2$ but a different parity distribution compared with the axion insulator. Therefore, the full indicator group of inversion symmetry $\mathbb{Z}_4 \times \mathbb{Z}_2 \times \mathbb{Z}_2 \times \mathbb{Z}_2$ is required to further distinguish these two phases⁴⁸, where the \mathbb{Z}_2 indicators can be chosen as the parity of the Chern number in the $k_i = \pi$ ($i = x, y, z$) plane. As shown in Fig. 1, the full symmetry indicator of the axion insulator phase and the 3D Chern insulator phase is $2:(000)$ and $2:(111)$, respectively.”*

REVIEWER COMMENTS

Reviewer #1 (Remarks to the Author):

The authors have satisfactorily addressed all of the comments. The manuscript has been significantly improved.

Reviewer #2 (Remarks to the Author):

The authors have made great efforts to improve the manuscript in their revised version and now provide a pretty compelling argument about why their results are different from those in the previous literature, especially Ref. 27 (now as Ref. 26). In the reply letter, they have also addressed most of my previous concerns, especially about the novelty of this work. Therefore, I can recommend this work for publication in Nature Communications after the authors address the following questions:

1) I still don't quite get why the in-band chiral hinge state "must" be there for the magnetic axion insulators. In the second paragraph of the section "Spectral signatures of the surface AHC – hinge states", the authors tried to explain the origin of the in-band hinge state with the statement "This can be understood by the chiral top surface AHC embedded into the helical gapless side surface states." Frankly speaking, I don't understand this statement at all.

Since the in-band hinge state appears as one of the major results in this work, I would like to ask the authors to clarify if the in-band hinge physics here is a phenomenon they merely observed or something they can actually understand. If it is the latter case, I would suggest the authors further elaborate on the topological origin of these in-band hinge states in the main text, which I believe is necessary to further enhance the novelty of this work. At least for now, this story in the manuscript is not very clear to me. Besides, I am also wondering if such in-band hinge state could be a generic phenomenon that works for all axion insulators.

2) In terms of the detection of the in-band hinge state, does the proposed transport experiment depend on the explicit orientation of the side surface? For example, what if the side surface has some surface defects?

3) Regarding the 3D QSH, I still think it would be good to actually show its explicit band representation in the main text, which will definitely help clarify the fragile nature of this phase.

Reviewer #3 (Remarks to the Author):

My questions in the first round have been answered satisfactorily. The authors have proposed clearer experimental predictions. For the AFM phase, which has gapless side surfaces protected by the time-reversal followed by a half-translation, the authors showed that the in-band hinge states have large spectral weights in one direction while small spectral weights in the other direction, reflecting the chiral nature. In the FM phase, the authors numerically showed the presence of chiral hinge modes surrounding the top surface. Such hinge modes can be distinguished from a trivial case where a 2D Chern insulator sits on top of a trivial insulator: In the axion case, if one flip spin polarization by applying a magnetic field, the hinge mode will be moved to the bottom surface; while in the latter, the position of the hinge modes will not change with a magnetic field. I recommend this work be published in Nature Communications.

One additional suggestion: For the AFM phase, the authors discussed the combined symmetry consisting of the time-reversal and the half-translation. It is this symmetry that protects the gapless side surface. It would be helpful if the authors can write down the explicit form of this symmetry below the Hamiltonian.

Comments of Referee #1 and authors' reply:

1.1 Referee: The authors have satisfactorily addressed all of the comments. The manuscript has been significantly improved.

Reply: Thank you.

Comments of Referee #2 and authors' reply:

2.1 Referee: The authors have made great efforts to improve the manuscript in their revised version and now provide a pretty compelling argument about why their results are different from those in the previous literature, especially Ref. 27 (now as Ref. 26). In the reply letter, they have also addressed most of my previous concerns, especially about the novelty of this work. Therefore, I can recommend this work for publication in Nature Communications after the authors address the following questions:

Reply: We thank the referee for appreciating the novelty of the work and the recommendation.

2.2 Referee: 1) I still don't quite get why the in-band chiral hinge state "must" be there for the magnetic axion insulators. In the second paragraph of the section "Spectral signatures of the surface AHC – hinge states", the authors tried to explain the origin of the in-band hinge state with the statement "This can be understood by the chiral top surface AHC embedded into the helical gapless side surface states." Frankly speaking, I don't understand this statement at all.

Since the in-band hinge state appears as one of the major results in this work, I would like to ask the authors to clarify if the in-band hinge physics here is a phenomenon they merely observed or something they can actually understand. If it is the latter case, I would suggest the authors further elaborate on the topological origin of these in-band hinge states in the main text, which I believe is necessary to further enhance the novelty of this work. At least for now, this story in the manuscript is not very clear to me. Besides, I am also wondering if such in-band hinge state could be a generic phenomenon that works for all axion insulators.

Reply: We thank the referee for asking this important question. The in-band hinge states essentially arise from the difference of the surface anomalous Hall conductivities of the two surfaces linked by the hinge. Let us consider the situation of two T -broken surfaces connected by

a hinge and the Fermi level is fixed at E_F cutting through the surface states for at least one surface. Then one can calculate the surface anomalous Hall conductivities for both surfaces, which are denoted by σ_1 and σ_2 [see schematic illustration in Fig. R1(a)]. If an electric field E_y is applied along the y direction, the anomalous Hall current densities are generated at both surfaces with $j_1 = \sigma_1 E_y$ and $j_2 = \sigma_2 E_y$, changing from j_1 to j_2 as the current passes through the hinge. Since the current density must be conserved, the hinge has to carry the extra current density $j_1 - j_2$. Thus, there has to be chiral modes localized at the hinge carrying the extra anomalous Hall current density from the surface, which means that the spectral weights of the left-moving modes and right-moving modes at the hinge have to be unequal. This gives rise to the chiral in-band hinge modes. A special situation is that one of the two surfaces is gapless and the other surface is gapped, then no matter where E_F is, the chiral hinge mode as a signature of the difference of the surface anomalous Hall effects, has to be embedded into the gapless surface state, which is exactly the case of the hinge mode in AFM MnBi_2Te_4 [Figs. 3(b) and 3(c)].

When E_F is in the gap of the surface states at both surfaces, the surface anomalous Hall conductivity is proportional to the bulk orbital magnetoelectric coupling, which is half quantized in axion insulators. Then one has to distinguish the case that the two surfaces have opposite surface AHCs $\pm 1/2$ or the same AHCs, in the former there is a chiral in-gap hinge mode [② in Fig. 3(f)], and in the latter there is no hinge mode [④ in Fig. 3(f)].

To better clarify the origin of the in-band hinge states, we have added the above discussion to Supplementary Note 6 and modified the relevant sentences at Page 9, Paragraph 1 of the main text to: *“The terminated AFM MnBi_2Te_4 sample and the calculated LDOS of the top hinge and side surface states are shown in Fig. 3(a-c) (position ② and ③). Compared with the helical gapless side surface states, we find that a remarkable feature of the hinge state is the asymmetric spectral weight between the left and right-moving modes, indicating its chiral nature. We denote such chiral hinge modes embedded inside the side surface bands as “in-band hinge” states, and attribute them to the difference of the surface anomalous Hall conductivities of the top and the side surfaces (see Supplementary Note 6 for details).”*

Since we believe that the in-band hinge state is the spectral signature of the surface anomalous Hall effect, we expect it as a generic phenomenon for axion insulators. For example, we have tested it on another axion insulator $\text{MnBi}_8\text{Te}_{13}$, and observed similar chiral hinge modes as shown in Fig. R1(b). To further illustrate that such chiral in-band hinge mode exists solely in magnetic axion insulators, we also provide a spectral comparison between an FM axion insulator (MnBi_2Te_4) and an FM trivial insulator (MnSb_2Te_4) in Supplementary Note 7.

Figure R1. (a) Schematic plot of the in-band hinge modes arise from the difference of the surface anomalous Hall conductivities of the two connecting surfaces. (b) Hinge states in FM axion insulator $\text{MnBi}_8\text{Te}_{13}$. It shows similar in-band and in-gap hinge states as reported in the current study of MnBi_2Te_4 .

2.3 Referee: 2) *In terms of the detection of the in-band hinge state, does the proposed transport experiment depend on the explicit orientation of the side surface? For example, what if the side surface has some surface defects?*

Reply: The in-band hinge states are considered to be robust for different orientations of the side surfaces. The chirality of the hinge state for a specific hinge is determined by the chirality of the surface anomalous Hall effect, which is reflected in our current data. For example, if the sample shown in Fig. 3(a) has inversion symmetry, the bottom-right hinge (point ④) should be identical to the top-left hinge, which can be regarded as a different side surface orientation compared to the top-right hinge (point ②). Note that Fig. 4(g) and Fig. 4(i) can also represent the hinge state between the top surface and the side surfaces with two different orientations.

Since the in-band hinge state is the signature of the surface anomalous Hall effect of the top/bottom surface, we do not expect qualitative change upon side surface defects. However, the in-gap hinge state may appear or disappear if the side surface defects are magnetic and reverse the sign of the local Chern marker of the side surface. Furthermore, according to Ref. 60, the transport signature of the in-band hinge state is robust against weak disorder.

2.4 Referee: 3) Regarding the 3D QSH, I still think it would be good to actually show its explicit band representation in the main text, which will definitely help clarify the fragile nature of this phase.

Reply: To check the fragile nature of the 3D T -broken QSH phase, we construct the elementary band representations (EBRs), i.e., the Wannier representations via putting p -orbital at different sites: A(0,0,0), B(1/2,0,0), C(0,1/2,0), D(1/2,1/2,0), E(0,0,1/2), F(1/2,0,1/2), G(0,1/2,1/2), and H(1/2,1/2,1/2), as shown in Supplementary Fig. 1(a). For each EBR, the parity distribution at inversion-invariant k -points is directly found. For instance, the parity $\pi_{\mathbf{k}}$ of the EBR $p@B$ is $\pi_{\Gamma} = \pi_Y = \pi_Z = \pi_V = +1$, and $\pi_X = \pi_M = \pi_U = \pi_W = -1$ [See Supplementary Fig. 1(b) for the definition of the high-symmetric k -points], where the p -orbital is invariant under $T_x P$ operation with T_x the translation operators along x direction.

As shown in Fig. 1 in the main text, the parity distribution of the 3D T -broken QSH phase is $(\pi_{\Gamma}, \pi_X, \pi_Y, \pi_M, \pi_U, \pi_V, \pi_W, \pi_Z) = (-2, +2, +2, +2, -2, +2, +2, +2)$. Consequently, one can decompose the 3D T -broken QSH state as $p@B + p@C + p@D - p@A$. In other words, such a representation does not admit any Wannier representation unless an additional trivial representation $p@A$ is considered together, as sketched in Supplementary Fig. 1(d).

In order to attract the audience's focus to the axion insulator phase, we prefer not to put the detailed discussion of the fragile nature of this phase in the main text. Instead, we put the relevant discussion to Supplementary Note 2.

Supplementary Figure 1. Construction of the elementary band representations. (a) Eight sites in a three-dimensional cell with inversion symmetry. (b) The Brillouin zone with eight inversion-invariant k -points. (c) Eight elementary band representations are constructed with a p -orbital state at one of the eight sites of (a). (d) The decomposition of the 3D T -broken QSH phase adding a trivial $p@A$ representation.

Comments of Referee #3 and authors' reply:

3.1 Referee: My questions in the first round have been answered satisfactorily. The authors have proposed clearer experimental predictions. For the AFM phase, which has gapless side surfaces protected by the time-reversal followed by a half-translation, the authors showed that the in-band hinge states have large spectral weights in one direction while small spectral weights in the other direction, reflecting the chiral nature. In the FM phase, the authors numerically showed the presence of chiral hinge modes surrounding the top surface. Such hinge modes can be distinguished from a trivial case where a 2D Chern insulator sits on top of a trivial insulator: In the axion case, if one flip spin polarization by applying a magnetic field, the hinge mode will be moved to the bottom surface; while in the latter, the position of the hinge modes will not change with a magnetic field. I recommend this work be published in Nature Communications.

Reply: Thank you.

3.2 Referee: One additional suggestion: For the AFM phase, the authors discussed the combined symmetry consisting of the time-reversal and the half-translation. It is this symmetry that protects the gapless side surface. It would be helpful if the authors can write down the explicit form of this symmetry below the Hamiltonian.

Reply: Done. We have changed the relevant sentence at Page 8, Paragraph 4 to “a gapless Dirac cone occurs due to the combined symmetry $T\tau_{1/2}$, where $\tau_{1/2}$ denotes the half-cell translation along the stacking axis⁵⁶”

REVIEWERS' COMMENTS

Reviewer #2 (Remarks to the Author):

In both the reply letter and the revised manuscript, the authors have satisfactorily addressed all my concerns and questions. Therefore, I can now recommend this work for publication in Nature Communications.